# E2Former-V2: On-the-Fly Equivariant Attention with Linear Activation Memory

Lin Huang [* 1]   Chengxiang Huang [* 1 2]   Ziang Wang [1]   Yiyue Du [3]   Chu Wang [1 4]   Haocheng Lu [1 5]   Yunyang Li [6]
Xiaoli Liu [1]   Arthur Jiang [1]   Jia Zhang [1]

## Abstract

Equivariant Graph Neural Networks (EGNNs) have become a widely used approach for modeling 3D atomistic systems. However, mainstream architectures face critical efficiency bottlenecks due to the explicit construction of geometric features or dense tensor products on *every* edge. To overcome this, we introduce **E2Former-V2**, an efficient architecture that integrates algebraic sparsity with hardware-aware execution. We introduce **Equivariant Axis-Aligned Sparsification (EAAS)**, which leverages an $SO(3) \rightarrow SO(2)$ change of basis to convert dense Wigner-$6j$ tensor contractions into sparse, parity-based re-indexing operations. Building on this representation, we propose **On-the-Fly Equivariant Attention**, a fully node-centric mechanism implemented via a fused Triton kernel. By eliminating materialized edge tensors and maximizing SRAM utilization, our kernel achieves up to **20× higher TFLOPS** than standard implementations. Experiments on SPICE and OMol25 show that E2Former-V2 preserves predictive accuracy while substantially accelerating inference, demonstrating the practical feasibility of large equivariant transformers on commodity GPUs. Our released code can be found at https://github.com/IQuestLab/UBio-MolFM/tree/main

## 1. Introduction

Machine learning techniques have gained increasing popularity in atomistic modeling (Bartók et al., 2013; 2010; Drautz, 2019). These methods offer significantly higher efficiency compared to quantum mechanical approaches, such

[1]IQuest Research Ubio Team [2]UNSW Sydney [3]Tsinghua University [4]Zhejiang University [5]Peking University [6]Yale University. Correspondence to: Jia Zhang <jialrs.z@iquestlab.com>.

*Proceedings of the 43$^{rd}$ International Conference on Machine Learning*, Seoul, South Korea. PMLR 306, 2026. Copyright 2026 by the author(s).

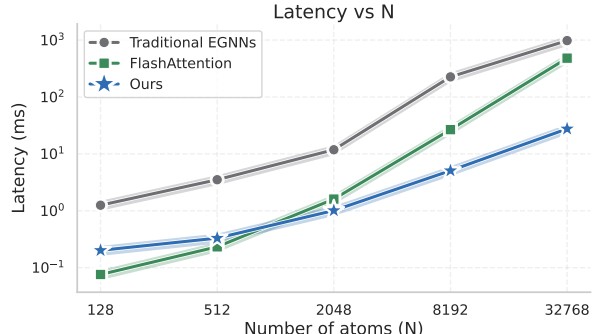

*Figure 1.* **Latency vs. number of atoms ($N$) for Traditional EGNNs, FlashAttention, and Ours.** FlashAttention consistently improves over Traditional EGNNs across all system sizes. The advantage of our method becomes increasingly pronounced as $N$ grows. See Appendix A for detailed experimental settings.

as Density Functional Theory (DFT) (Hohenberg & Kohn, 1964; Kohn & Sham, 1965), while maintaining comparable performance. In this domain, *equivariant graph neural networks* have emerged as a promising class of models, as they preserve the rotational and translational symmetries required for physically meaningful predictions (Schütt et al., 2018; Gasteiger et al., 2024; Geiger et al., 2022; Unke et al., 2021; Liao & Smidt, 2023).

Recent architectures such as MACE (Batatia et al., 2022; Kovács et al., 2025), Allegro(Musaelian et al., 2022), ViS-Net (Wang et al., 2023), eSCN (Passaro & Zitnick, 2023), EquiformerV2 (Liao et al., 2023), GotenNet (Aykent & Xia, 2025), eSEN (Fu et al., 2025b), E2Former (Li et al., 2025) , and UMA (Wood et al., 2025) use rich geometric representations and expressive message passing. These design choices lead to notable performance improvements. Despite their differing mathematical formulations, these models share a common characteristic: **their computation and memory footprint are inherently edge-centric.** In practice, they explicitly construct geometric features or perform tensor products on *every* edge. This results in $\mathcal{O}(|\mathcal{E}|) = \mathcal{O}(kN)$ memory and activation costs for molecular graphs with bounded degree ($k \approx 30$–$100$).

To quantify the impact of this design, we conducted an

observational analysis comparing these traditional EGNNs against standard attention mechanisms. As illustrated by the gray line in Figure 1, traditional EGNNs suffer from severe latency due to their edge-centric nature. In contrast, standard Transformers have overcome similar bottlenecks through *hardware-aligned execution* and SRAM-optimized kernels (Dao et al., 2022; Dao, 2023; Shah et al., 2024). FlashAttention (Dao et al., 2022), represented by the green line in Figure 1, computes attention in a streaming, tile-based manner. This design avoids storing the full attention matrix in off-chip memory. As a result, the activation memory scales as $\mathcal{O}(N)$ rather than $\mathcal{O}(N^2)$, bridging the gap between GPU compute throughput and memory bandwidth.

To the best of our knowledge, these execution principles remain unexplored in existing $SO(3)$-equivariant architectures. This poses a fundamental question: *can equivariant attention be reformulated to support SRAM-optimized, streaming execution?* Such a formulation would eliminate explicit edge activations, ensuring memory complexity remains linear with the number of atoms. We demonstrate that this is achievable with **E2Former-V2**, a fully node-centric architecture that brings FlashAttention-style streaming computation to equivariant models. As shown by the blue line in Figure 1, our approach significantly alleviates the aforementioned bottlenecks. Significantly, this performance gain widens as $N$ increases, underscoring E2Former-V2's efficacy in handling large-scale atomic structures that were previously computationally prohibitive.

E2Former-V2 achieves $\mathcal{O}(|\mathcal{V}|)$ activation memory while preserving theoretical exactness through two core designs: algebraic sparsification in an $SO(2)$ representation space and a fully fused, node-centric equivariant attention kernel. Our contributions are summarized as follows:

1. **Equivariant Axis-Aligned Sparsification (EAAS).** By unifying Wigner-$6j$ recoupling with $SO(2)$ axis alignment, EAAS transforms dense $SO(3)$ tensor contractions into sparse parity re-indexing operations, yielding a $\sim 1.5\times$ speedup in the convolution stage.

2. **On-the-Fly Equivariant Attention.** We introduce a fully fused Triton kernel that avoids $\mathcal{O}(|\mathcal{E}|)$ edge materialization. By maximizing on-chip SRAM reuse, it eliminates the dominant memory bottleneck and achieves up to **20×** higher TFLOPS than standard implementations.

3. **Scalable performance.** Extensive experiments on SPICE and OMol25 show that E2Former-V2 preserves predictive accuracy while significantly improving training throughput and memory efficiency over prior equivariant models.

4. **Physically faithful MD.** E2Former-V2 reproduces experimental liquid water structure and enables stable molecular dynamics refinement of AlphaFold-predicted protein structures ($\sim$30k atoms).

## 2. Related Work

**Equivariant graph neural networks.** Equivariant graph neural networks (GNNs) have become a central paradigm for modeling 3D geometric data. Early works emphasized computational efficiency by restricting representations to scalars and vectors ($L \leq 1$). Methods such as EGNN (Satorras et al., 2022), PaiNN (Schütt et al., 2021), and ViSNet (Wang et al., 2023) avoid explicit tensor products, achieving linear complexity in graph size. While efficient, this restriction limits their ability to model higher-order angular correlations encoded in spherical harmonics.

To improve expressivity, Tensor Field Networks (TFN) (Thomas et al., 2018), Allegro (Musaelian et al., 2022) and NequIP (Batzner et al., 2022) introduced full $SO(3)$-equivariant convolutions based on tensor products. Concretely, these models encode relative geometry with spherical harmonics and couple irreps across angular orders via Clebsch–Gordan tensor products, enabling rich anisotropic interactions. However, repeated irrep coupling and tensor-product expansions are costly, scaling as $O(L^6)$. Subsequent works, including eSCN (Passaro & Zitnick, 2023), EquiformerV2 (Liao et al., 2023), and eSEN (Fu et al., 2025a) reduce this complexity to $O(L^3)$ by exploiting a change of basis to $SO(2)$. Despite these advances, most existing approaches remain *edge-centric*, relying on the explicit construction and storage of edge-wise features. Recent studies have begun to question whether such edge materialization is fundamentally necessary. E2Former (Li et al., 2025) provides an important theoretical foundation by demonstrating the mathematical feasibility of this factorization, which reduces the *theoretical* complexity from $\mathcal{O}(|\mathcal{E}|)$ to $\mathcal{O}(|\mathcal{V}|)$. However, reductions in arithmetic complexity alone do not guarantee practical speedups, as memory access patterns and hardware constraints are largely ignored.

**Efficient Equivariant Operations.** Despite their data efficiency, equivariant models are often limited by the computational complexity of tensor products (TP). Recent studies have sought to mitigate this from both theoretical and system perspectives: Xie et al. (2025) characterize the expressivity-runtime trade-offs in TP architectures, while Luo et al. (2024) propose Gaunt tensor products for efficient operations in the Fourier basis. Further, Lin et al. (2024) leverage tensor decomposition to accelerate interatomic potential computations. On the systems level, highly optimized libraries such as cuEquivariance NVIDIA (2024) and OpenEquivariance Bharadwaj et al. (2025) have been developed to provide high-performance primitives for equivariant operations. Our work builds on these efforts by introducing a fused kernel approach that significantly reduces memory and runtime overhead in large-scale

simulations.

**Hardware-Aware Efficient Deep Learning.** This gap becomes particularly evident on modern GPU architectures, where performance is increasingly dominated by memory access rather than floating-point computation (Wulf & McKee, 1995; Williams et al., 2009). Although high-bandwidth memory (HBM) provides large capacity, its bandwidth remains orders of magnitude lower than that of on-chip SRAM (Jia et al., 2018), causing many deep learning workloads to be constrained by data movement rather than computation (Ivanov et al., 2021). Recent advances in Transformer models demonstrate that hardware-aware execution can fundamentally alleviate such memory bottlenecks. Methods such as FlashAttention (Dao et al., 2022; Shah et al., 2024) eliminate materialized attention matrices through kernel fusion and tiling, keeping intermediate results in SRAM and significantly reducing memory I/O.

Despite their success in Transformers, extending such hardware-aware execution strategies to equivariant GNNs remains largely unexplored. The irregular structure of graphs and the algebraic complexity of spherical harmonic operations introduce unique challenges, creating a gap between algorithmic factorization and hardware-efficient realization.

## 3. Preliminaries

We introduce the notation and mathematical background of SO(3)-equivariant graph neural networks, and briefly review the node-centric factorization proposed in E2Former (Li et al., 2025), which serves as the theoretical foundation of our method.

**Graph and representations.** We consider a molecular graph $\mathcal{G} = (\mathcal{V}, \mathcal{E})$ with $N = |\mathcal{V}|$ nodes. Each node $i$ is associated with a 3D coordinate $\vec{r}_i \in \mathbb{R}^3$ and a feature representation $\mathbf{h}_i$. Relative positions are defined as $\vec{r}_{ij} = \vec{r}_j - \vec{r}_i$.

Node features are modeled as a direct sum of irreducible representations (irreps) of SO(3):

$$\mathbf{h}_i = \bigoplus_{\ell=0}^{L} \mathbf{h}_i^{(\ell)}, \quad \mathbf{h}_i^{(\ell)} \in \mathbb{R}^{C_\ell \times (2\ell+1)}. \quad (1)$$

Under a rotation $R \in$ SO(3), features of degree $\ell$ transform as $\mathbf{h}_i^{(\ell)} \mapsto D^{(\ell)}(R)\mathbf{h}_i^{(\ell)}$, where $D^{(\ell)}$ denotes the Wigner-$D$ matrix.

**Geometric encoding.** Geometric information is encoded using *solid spherical harmonics* $\mathcal{R}^{(\ell)}(\vec{r}) \in \mathbb{R}^{2\ell+1}$, defined as

$$\mathcal{R}_m^{(\ell)}(\vec{r}) = \|\vec{r}\|^\ell Y_{\ell,m}\left(\frac{\vec{r}}{\|\vec{r}\|}\right), \quad (2)$$

where $Y_{\ell,m}$ are real spherical harmonics.

**Equivariant tensor product.** We denote by $\otimes$ the Clebsch–Gordan tensor product, which is a bilinear map between irreducible representations of SO(3). In the following derivation, we omit the parity for notational simplicity. Given two irreps $\mathbf{u}^{(\ell_1)}$ and $\mathbf{v}^{(\ell_2)}$, their coupling to an output irrep of degree $\ell_{\text{out}}$ is defined as:

$$\left(\mathbf{u}^{(\ell_1)} \otimes \mathbf{v}^{(\ell_2)}\right)_m^{(\ell_{\text{out}})} = \sum_{m_1,m_2} C_{\ell_1,m_1;\ell_2,m_2}^{\ell_{\text{out}},m} u_{m_1}^{(\ell_1)} v_{m_2}^{(\ell_2)}, \quad (3)$$

where $C_{\cdots}^{\cdots}$ are the Clebsch–Gordan coefficients.

**E2Former: node-centric factorization.** E2Former (Li et al., 2025) reformulates spherical convolution by algebraically factorizing the tensor product between node features and relative geometric encodings. Specifically, it considers an attention-weighted aggregation of edge messages:

$$\mathbf{m}_i = \sum_{j \in \mathcal{N}(i)} \alpha_{ij} \mathbf{m}_{ij}, \quad (4)$$

where $\alpha_{ij}$ is a scalar coefficient and each edge message is defined as $\mathbf{m}_{ij} = \mathbf{h}_j \otimes \mathcal{R}^{(\ell)}(\vec{r}_{ij})$.

Using the Binomial Local Expansion theorem together with Wigner-$6j$ recoupling, E2Former shows that the tensor product involving the relative position $\vec{r}_{ij} = \vec{r}_j - \vec{r}_i$ can be decomposed into a sum of terms that separately depend on the source node $j$ and the target node $i$. The resulting factorized form is given by:

$$\mathbf{m}_i^{(\ell_{\text{out}})} = \sum_{u=0}^{\ell} (-1)^{\ell-u} \binom{\ell}{u} \Bigg[ \underbrace{\mathcal{R}^{(u)}(\vec{r}_i)}_{\text{Target Node}} \otimes_{6j} \\ \left( \sum_{j \in \mathcal{N}(i)} \alpha_{ij} \cdot \underbrace{\left(\mathbf{h}_j \otimes \mathcal{R}^{(\ell-u)}(\vec{r}_j)\right)}_{\text{Source Node}} \right) \Bigg]^{(\ell_{\text{out}})}. \quad (5)$$

In Eq. (5), the inner summation aggregates source-node features $\mathbf{h}_j$ coupled with their absolute geometric encodings $\mathcal{R}^{(\ell-u)}(\vec{r}_j)$. The outer tensor product combines this aggregated source representation with a target-node-dependent geometric factor $\mathcal{R}^{(u)}(\vec{r}_i)$. The operator $\otimes_{6j}$ denotes an SO(3)-equivariant recoupling where the path weight is parameterized by Wigner-$6j$ recoupling coefficients.

## 4. Method

### 4.1. Motivation for E2Former-V2.

Despite the theoretical elegance of Eq. (5), practical implementation encounters two bottlenecks. **Firstly**, the equivariant tensor contractions, including both standard CG-products ($\otimes$) and their Wigner-$6j$ counterparts ($\otimes_{6j}$), require dense summation over Clebsch–Gordan coupling

channels and incur $O(L^6)$ computational complexity. **Secondly**, while E2Former achieves a conceptual decoupling at the level of equivariant convolution operators, the execution remains edge-centric. both scalar attention weight $\alpha_{ij}$ and source node message $\Big(\sum_{j\in\mathcal{N}(i)}\alpha_{ij}\cdot$

$\underbrace{\Big(\mathbf{h}_j\otimes\mathcal{R}^{(\ell-u)}(\vec{r}_j)\Big)}_{\text{Source Node}}\Big)$ remains edge-dependent. In standard implementations based on automatic differentiation frameworks, these edge-wise quantities are materialized as $\mathcal{O}(|\mathcal{E}|)$ tensors in High Bandwidth Memory (HBM), causing the computation to become bandwidth-bound and reintroducing a memory wall for large-scale systems.

We propose **E2Former-V2**, a unified equivariant architecture designed to realize the theoretical promise of linear-scaling learning with respect to the number of nodes. Existing realizations remain constrained by two fundamental bottlenecks: the *arithmetic intensity* of **dense** SO(3) **tensor contractions** and the *memory wall* of explicit edge instantiations. Our method overcomes these barriers through a holistic design (Fig. 2) that combines an algebraic SO(3) → SO(2) reduction with a hardware-aware, SRAM-efficient, on-chip execution model.

### 4.2. Node-Centric equivariant attention.

Our attention mechanism serves as the architectural realization of the node-centric factorization derived in Section 3. Rather than materializing high-dimensional edge features as first-class tensors, we design a pipeline in which attention weights depend only on SO(3)-invariant scalar quantities, while equivariant geometric information is propagated through the node-wise value path.

**Equivariant feature projection.** To preserve SO(3) equivariance, all learnable transformations are required to commute with the group action. By Schur's lemma(Serre et al., 1977), this restricts linear maps to act independently within each irreducible $\ell$-subspace. Accordingly, each node $i$ carries equivariant irrep features $\mathbf{h}_i^{(\ell)}\in\mathbb{R}^{h_\ell\times(2\ell+1)}$.

Equation (6) constructs query and key representations by first applying $\ell$-wise equivariant linear projections and then performing an SO(3)-invariant inner product within each angular order. Here $W_{Q1}^{(\ell)}, W_{Q2}^{(\ell)}, W_{K1}^{(\ell)}, W_{K2}^{(\ell)}\in\mathbb{R}^{h_\ell\times h_\ell}$ act on the multiplicity dimension only. Equivalently, the overall projections are block-diagonal with respect to the $\ell$ decomposition and therefore commute with the SO(3) representation.

$$\mathbf{q}_i = \mathrm{concat}_{\ell=0}^{L}\Big(\big\langle W_{Q1}^{(\ell)}\mathbf{h}_i^{(\ell)}\ ,\ W_{Q2}^{(\ell)}\mathbf{h}_i^{(\ell)}\big\rangle\Big),$$
$$\mathbf{k}_j = \mathrm{concat}_{\ell=0}^{L}\Big(\big\langle W_{K1}^{(\ell)}\mathbf{h}_j^{(\ell)}\ ,\ W_{K2}^{(\ell)}\mathbf{h}_j^{(\ell)}\big\rangle\Big), \tag{6}$$

Here concat denotes concatenation across angular orders, and $\langle\cdot,\cdot\rangle$ denotes an SO(3)-invariant inner product defined by contraction over the magnetic quantum number:

$$\langle\mathbf{h}^{(\ell)},\mathbf{h}'^{(\ell)}\rangle := \sum_{m=-\ell}^{\ell}\mathbf{h}_m^{(\ell)}\cdot\mathbf{h}_m'^{(\ell)},$$

which removes the $m$-dependence and yields invariant scalar features (per channel) for each $\ell$.

**Invariant attention scoring.** Given the per-$\ell$ invariant query and key vectors constructed in Eq. (6), we compute a scalar attention weight for each neighbor $j\in\mathcal{N}(i)$ via a distance-modulated dot product:

$$\alpha_{ij} = \mathrm{softmax}_{j\in\mathcal{N}(i)}\Big(\tfrac{1}{\sqrt{d_k}}\mathbf{q}_i^\top\mathbf{k}_j\ +\ b(r_{ij})\Big)\phi(r_{ij}), \tag{7}$$

where $b(r_{ij})$ and $\phi(r_{ij})$ are scalar functions of the interatomic distance (e.g., an RBF-based bias and a cutoff gate). Because $\mathbf{q}_i$ and $\mathbf{k}_j$ are SO(3)-invariant, the resulting score is rotationally invariant. Importantly, $\alpha_{ij}$ is a *scalar* edge weight (carrying no angular momentum) and is the only edge-dependent quantity used in the attention weighting, enabling a memory-light implementation. More detailed information can be found in Appendix D.

**Implementation of factorized message passing.** We realize the factorization theorem shown in Eq. 5 as a three-stage data flow. This design ensures that directional information enters only on the value path, strictly decoupling edge interactions:

(i) **Source-term preparation.** Before message passing, we pre-couple the value features $\mathbf{v}_j$ with local spherical harmonics at the source node. This operation depends solely on node $j$:

$$\mathbf{h}_j' = \mathbf{h}_j\otimes\mathcal{R}(\vec{r}_j). \tag{8}$$

(ii) **Weighted aggregation.** We transmit these pre-computed terms using the scalar attention weights $\alpha_{ij}$. We define the aggregated spatial message $\mathbf{m}_i$ as the weighted sum of neighbor source terms:

$$\mathbf{m}_i = \sum_{j\in\mathcal{N}(i)}\alpha_{ij}\cdot\mathbf{h}_j'. \tag{9}$$

Unlike standard messages, $\mathbf{m}_i$ aggregates geometric information without yet resolving the target orientation.

(iii) **Target-term coupling.** Finally, the aggregated message $\mathbf{m}_i$ is coupled with the target node's spherical harmonics $\mathcal{R}(\vec{r}_i)$ to recover the full equivariant update $\hat{\mathbf{h}}_i$:

$$\hat{\mathbf{h}}_i = \mathbf{m}_i\otimes\mathcal{R}(\vec{r}_i). \tag{10}$$

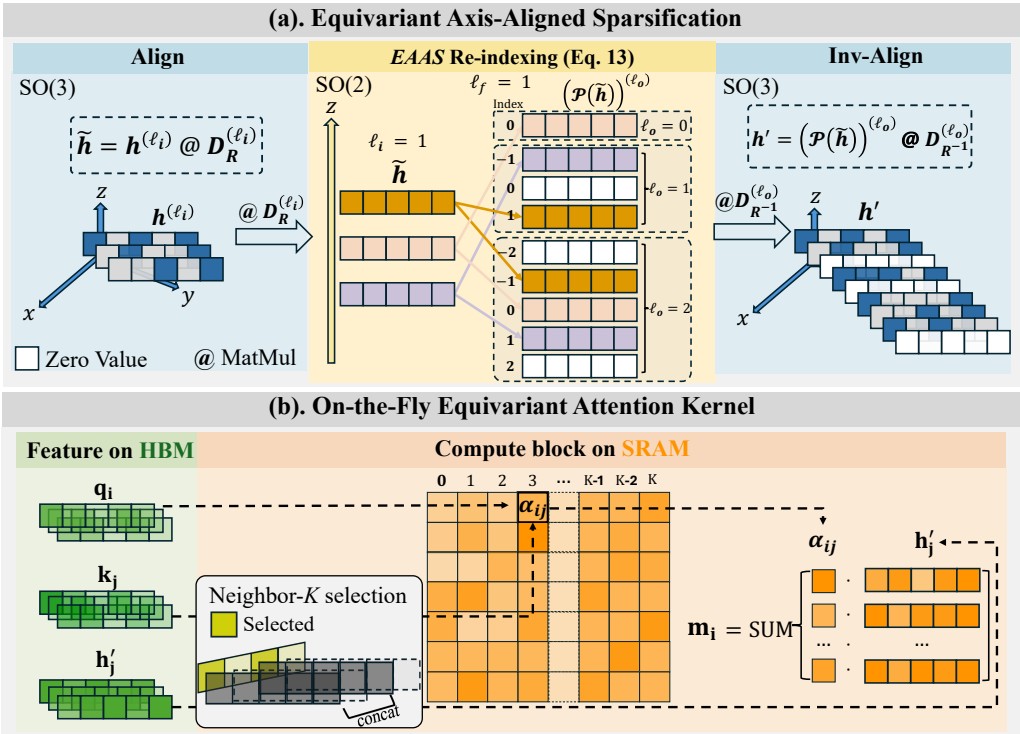

*Figure 2.* **Key components of E2Former-V2.** (a) **EAAS.** E2Former-V2 aligns features with $D_R$, applies the sparse EAAS re-indexing operator $\mathcal{P}$ (Eq. 13) in the axis-aligned frame, and inverse-aligns with $D_{R^{-1}}$. The visualization shows the re-indexing pattern for $\ell_i = 1$ and $\ell_f = 1$. (b) **On-the-fly equivariant attention.** E2Former-V2 computes attention by streaming over neighbors and accumulating the output on the fly, avoiding explicit materialization of edge-level intermediates.

### 4.3. Equivariant Axis-Aligned Sparsification (EAAS)

In practice, the dominant cost in node-centric architectures arises from a critical algebraic bottleneck: dense $\mathrm{SO}(3)$ tensor products exhibit highly irregular contraction patterns. Their complex index structure and selection rules lead to irregular memory access patterns and poor SIMD utilization, making them ill-suited for efficient execution on modern GPUs. Equivariant Axis-Aligned Sparsification (EAAS) addresses this mismatch by reformulating dense $\mathrm{SO}(3)$ tensor products into sparse, permutation-based operations with regular memory access patterns, thereby aligning equivariant computation with GPU-friendly execution.

At a high level, EAAS exploits the fact that geometric encodings become maximally sparse when expressed in a local, axis-aligned frame. By commuting rotations through the tensor product, dense couplings collapse into a fixed re-indexing rule followed by lightweight blockwise linear maps.

**Lemma 4.1** (Pole Sparsity of Solid Spherical Harmonics). *Let $R \in \mathrm{SO}(3)$ be a rotation that aligns the global z-axis with a vector $\vec{r}$. Then the solid spherical harmonics satisfy*

$$\mathcal{R}_m^{(\ell)}(R\vec{r}) \propto \delta_{m,0}. \tag{11}$$

Lemma 4.1 implies that geometric features become maximally sparse in the aligned frame, providing the key ingredient for eliminating dense summation over magnetic indices. Motivated by this observation, we introduce an alignment rotation $R$ and its corresponding representation matrices $D_R^{(\ell)}$, which map features from the global $\mathrm{SO}(3)$ frame into the local axis-aligned frame. We write the aligned node features as

$$\tilde{h} := h^{(\ell_i)} @ D_R^{(\ell_i)}, \tag{12}$$

where @ denotes matrix multiplication.

**Definition 4.2** (EAAS Re-indexing Operator). Let $\tilde{h}$ denote node features expressed in the aligned frame. We define the *re-indexing operator* $\mathcal{P}$ as the blockwise linear map (acting independently on each degree $\ell$) whose componentwise action is given by

$$\left(\mathcal{P}(\tilde{h})\right)_{m_o}^{(\ell_o)} :=$$
$$\begin{cases} C_{(\ell_i,m_i),(\ell_f,0)}^{(\ell_o,m_o)} \tilde{h}_{m_i}^{(\ell_i)}, & \text{if } L_\Sigma \text{ is even,} \\ -2(-1)^{m_o} C_{(\ell_i,-m_i),(\ell_f,0)}^{(\ell_o,m_o)} \tilde{h}_{-m_i}^{(\ell_i)}, & \text{if } L_\Sigma \text{ is odd,} \end{cases} \tag{13}$$

where $L_\Sigma = \ell_i + \ell_f + \ell_o$ and $m_i = m_o$.

The operator $\mathcal{P}$ therefore implements a deterministic re-indexing within each $\ell$-block, so that for each output order

$m_o$ at most one input order $m_i$ contributes. This yields a sparse, permutation-like operation and avoids explicitly materializing dense Clebsch–Gordan contraction paths. A complete derivation of the above rule from Clebsch–Gordan coefficients, including the parity selection and sign convention, is provided in Appendix C.

**Proposition 4.3** (Equivariant Axis-Aligned Sparsification). *Let $R \in \mathrm{SO}(3)$ align the z-axis with $\vec{r}$. The SO(3)-equivariant tensor product between node features $h^{(\ell_i)}$ and geometric encoding $\mathcal{R}^{(\ell_f)}(\vec{r})$ admits the exact form*

$$\left( h^{(\ell_i)} \otimes \mathcal{R}^{(\ell_f)}(\vec{r}) \right)^{(\ell_o)}_{m_o} = \left( \mathcal{P}(\tilde{h}) \right)^{(\ell_o)} @ D^{(\ell_o)}_{R^{-1}}, \quad (14)$$

*where the repeated index $m_i$ is implicitly summed over.*

Proposition 4.3 shows that dense SO(3) tensor products can be implemented via rotation conjugation and a sparse, blockwise re-indexing operator. A complete derivation of Proposition 4.3, including the parity selection rules and concrete low-degree examples, is given in Appendix C.

### 4.4. On-the-Fly Equivariant Attention

This section presents a fused GPU kernel for computing the equivariant attention aggregation defined in Eq. (9). The kernel is designed to eliminate the explicit materialization of edge-level intermediate tensors by evaluating sparse attention as a node-centric, streaming reduction over neighbors. We use $H$ to denote the number of attention heads. For clarity, we present the single-head formulation ($H = 1$) and omit the head index; the multi-head case follows by applying the same computation independently per head.

**Neighbor indexing and sparse gather.** We consider a molecular system with $N$ atoms. For each target atom $i$, let $\mathcal{N}(i) = \{j_1, \ldots, j_{K_i}\}$ denote its neighbor set, where $K_i \leq K$ and $K$ represents the maximum number of neighbors. The neighbor list induces implicit gather operations from node-indexed features to neighbor-indexed accesses. Given node-wise queries and keys $q, k \in \mathbb{R}^{N \times d}$ and source-term features $h' \in \mathbb{R}^{N \times C}$, accesses such as $k_j$ and $h'_j$ with $j \in \mathcal{N}(i)$ correspond to indirect memory reads. These gathers are resolved dynamically inside the kernel and are never materialized as dense tensors of shape $N \times K \times d$ or $N \times K \times C$.

**Streaming node-centric formulation.** The proposed kernel evaluates the aggregation in Eq. (9) as a streaming reduction. As shown in Algorithm 1, rather than computing all scores in advance, the kernel iterates over neighbors $j \in \mathcal{N}(i)$ and evaluates each inner product on the fly. To ensure numerical stability, we use an online softmax formulation. For each atom $i$, the kernel maintains a running maximum $\mu_i$, a normalization accumulator $z_i$, and a value accumulator $\mathbf{A}_i \in \mathbb{R}^C$:

$$\mu_i^{(k)} = \max\left( \mu_i^{(k-1)}, s_{ij} \right), \quad (15)$$

$$z_i^{(k)} = z_i^{(k-1)} \exp\left( \mu_i^{(k-1)} - \mu_i^{(k)} \right) \\ + \exp\left( s_{ij} - \mu_i^{(k)} \right), \quad (16)$$

$$\mathbf{A}_i^{(k)} = \mathbf{A}_i^{(k-1)} \exp\left( \mu_i^{(k-1)} - \mu_i^{(k)} \right) \\ + \exp\left( s_{ij} - \mu_i^{(k)} \right) \phi(r_{ij}) \mathbf{h}'_j, \quad (17)$$

where $j \in \mathcal{N}(i)$ and $\mathbf{h}'_j$ is the source-term feature for node $j$. The unnormalized score $s_{ij}$ is recalled from Eq. (7):

$$s_{ij} = \frac{1}{\sqrt{d_k}} q_i^\top k_j + b(r_{ij}). \quad (18)$$

After all neighbors are processed, the final aggregated message for the head is $m_i = \mathbf{A}_i^{(K)} / z_i^{(K)}$.

**Memory and performance implications.** By avoiding the explicit materialization of the attention scores $\alpha_{ij}$ and the gathered value tensors, the fused kernel eliminates the dominant sources of intermediate memory allocation and HBM traffic in sparse attention. All reductions over the neighbor dimension are performed on chip, and each key and value vector is loaded exactly once per interaction. As a result, the kernel substantially reduces intermediate memory allocation and HBM traffic in sparse attention, alleviating the memory bottleneck and improving arithmetic utilization on modern GPUs.

---

**Algorithm 1** Fused On-the-Fly Equivariant Attention (Forward Pass, $H = 1$)

---

**Require:** $q, k \in \mathbb{R}^{N \times d}$, $h' \in \mathbb{R}^{N \times C}$, neighbor set $\mathcal{N} \in \mathbb{R}^{N \times K}$, bias $b(r)$, radial scaling $\phi(r)$, scale $\tau = 1/\sqrt{d_k}$
**Ensure:** Aggregated message $m \in \mathbb{R}^{N \times C}$
1: **for** each target atom $i \in \{1, \ldots, N\}$ **in parallel do**
2:    $\mu \leftarrow -\infty$, $z \leftarrow 0$, $\mathbf{A} \leftarrow \mathbf{0}$
3:    **for** $j \in \mathcal{N}(i)$ **do**
4:       **if** $j$ is padding **then**
5:          **continue**
6:       **end if**
7:       $s \leftarrow \tau \cdot q_i^\top k_j + b(r_{ij})$
8:       $\mu' \leftarrow \max(\mu, s)$
9:       $z \leftarrow z \cdot e^{\mu - \mu'} + e^{s - \mu'}$
10:      $\mathbf{A} \leftarrow \mathbf{A} \cdot e^{\mu - \mu'} + e^{s - \mu'} \cdot \phi(r_{ij}) h'_j$
11:      $\mu \leftarrow \mu'$
12:    **end for**
13:    $m_i \leftarrow \mathbf{A}/z$
14: **end for**

---

# 5. Experiments

We evaluate E2Former-V2 on standard molecular benchmarks to verify its computational efficiency, scalability, and predictive accuracy. Our experiments aim to confirm that the proposed EAAS and on-the-fly equivariant attention kernel reduce the computational complexity without compromising the model's expressivity.

## 5.1. Performance comparison with related methods.

We utilize **SPICE** (Eastman et al., 2022) to verify precision in medicinal chemistry contexts (e.g., protein-ligand interactions), and **OMol25** (Levine et al., 2025) to assess high-throughput capabilities across the massive, diverse chemical spaces required for foundation models. Details of the two datasets are provided in Appendix F.

To evaluate the expressivity and generalizability of E2Former-V2, we benchmark it against recent equivariant architectures, including MACE (Batatia et al., 2022), eSEN (Fu et al., 2025a), UMA (Wood et al., 2025), and E2Former-V1 (Li et al., 2025).

**Firstly**, we evaluate generalization on the SPICE dataset. As shown in Table 1, E2Former-V2 achieves the lowest Energy and Force MAE across most subsets, including challenging regimes such as *Monomers*, *Dimers*, and *Solvated Amino Acids*. On *Dimers*, it reduces the Energy MAE by 48% relative to MACE-Large, demonstrating that the EAAS-based $SO(2)$ formulation effectively captures high-order geometric interactions. **Secondly**, we assess scalability on the large-scale OMol25 dataset. As summarized in Table 2, E2Former-V2 remains competitive in this regime: the conservative variant achieves an aggregate Energy MAE of $1.27$ meV/atom, matching eSEN-small and significantly outperforming UMA-S ($3.62$ meV/atom), confirming its suitability as an efficient backbone for large-scale molecular foundation models.

## 5.2. Efficiency and Scalability Analysis

We validate the computational efficiency and scalability of E2Former-V2 from both algebraic and hardware perspectives.

**Direct Kernel Efficiency.** To ensure a rigorous comparison, we further benchmark EAAS against highly optimized backends, including `cuEquivariance` and `OpenEquivariance`, using the setting $(128{\times}0e + 128{\times}1e + 128{\times}2e) \otimes (1{\times}2e)$. As shown in Table 3, EAAS remains the fastest across the entire range, achieving a speedup of approximately $3.6\times$ to $6\times$ over e3nn and $1.5\times$ to $3.6\times$ over `OpenEquivariance`. EAAS maintains superior throughput across all regimes, confirming that our efficiency gains are structural rather than merely baseline-dependent.

**Secondly,** we assess the system-level performance of our fused Equivariant Flash Attention Kernel. Figure 3 compares the throughput (TFLOPS) and peak memory usage of our kernel against a naive PyTorch implementation. As shown in Figure 3a and Figure 3b, our method demonstrates superior scalability: the TFLOPS rapidly increase with the number of neighbors and atoms before saturating at a high utilization rate, whereas the naive implementation remains at a consistently low throughput level. Consequently, we achieve an approximate $20\times$ speedup. This trajectory indicates that our method effectively shifts the workload from being memory-bound to compute-bound as scale increases. Furthermore, as shown in Figure 3c and Figure 3d, our approach maintains a significantly lower memory footprint compared to the naive baseline, confirming its ability to scale efficiently to larger systems. A detailed comparison against dense FlashAttention on H20 is provided in Appendix J.

**Thirdly**, we perform an end-to-end component ablation on a conservative model to isolate the contribution of EAAS and the fused Triton kernel. Table 4 toggles each component independently. Both components contribute to the final speedup, but in different ways: the Triton fused kernel provides the dominant end-to-end gain because edge-level materialization is the primary bottleneck in both runtime and memory. In contrast, enabling EAAS alone cannot remove the main edge-level memory bottleneck, which explains why configurations without the fused kernel run into OOM at 10k atoms. The full system consistently achieves the best throughput (e.g. $1.40 \rightarrow 4.24$ samples/s at 5k atoms). Throughput at larger scales is reported in Table 5.

## 5.3. Comparison of inference speed.

To evaluate the inference efficiency of E2Former-V2, we benchmark it against a broad range of related architectures, including, eSEN (Fu et al., 2025a) , MACE (both OMOL and Large variants) (Batatia et al., 2022), E2Former-V1 (Li et al., 2025) , UMA-S (Wood et al., 2025), GotenNet (Aykent & Xia, 2025) , Allegro (Musaelian et al., 2022) , EscAIP (Qu & Krishnapriyan, 2024), and EquiformerV2 (Liao et al., 2023). As presented in Table 5, we measure the inference throughput (steps/s) across system sizes ranging from 1k to 100k atoms. **Firstly**, under memory constraints, prior equivariant Transformers and high-order potentials (e.g., EquiformerV2, E2Former-V1, MACE-Large) encounter OOM failures beyond 10k–50k atoms. In contrast, E2Former-V2 scales reliably to 100k atoms in both settings, indicating that our design removes the dominant memory bottlenecks. **Secondly**, E2Former-V2 also achieves the highest throughput across all scales. In the Conservative setting, it reaches 0.12 steps/s at 100k atoms, nearly $3\times$ faster than UMA-S. In the Direct setting, it attains 58.33 steps/s at 1k atoms—an order-of-magnitude faster than Alle-

*Table 1.* **Performance comparison on the SPICE dataset.** Results are reported in Energy ($E$, meV/atom) and Force ($F$, meV/Å) MAE. **Bold** and underline denote the best and second-best performers. Shaded rows indicate our proposed method.

| MODEL | PUBCHEM | | MONOMERS | | DIMERS | | DIPEPTIDES | | SOLV. AMINO | | WATER | | QMUGS | | ALL | |
|---|---|---|---|---|---|---|---|---|---|---|---|---|---|---|---|---|
| | $E$ | $F$ | $E$ | $F$ | $E$ | $F$ | $E$ | $F$ | $E$ | $F$ | $E$ | $F$ | $E$ | $F$ | $E$ | $F$ |
| MACE Small | 1.41 | 35.68 | 1.04 | 17.63 | 0.98 | 16.31 | 0.84 | 25.07 | 1.60 | 38.56 | 1.67 | 28.53 | 1.03 | 41.45 | 1.27 | 29.76 |
| MACE Medium | 0.91 | 20.57 | 0.63 | 9.36 | 0.58 | 9.02 | 0.52 | 14.27 | 1.21 | 23.26 | 0.76 | 15.27 | 0.69 | 23.58 | 0.80 | 17.03 |
| MACE Large | 0.88 | 14.75 | 0.59 | 6.58 | 0.54 | 6.62 | 0.42 | 10.19 | 0.98 | 19.43 | 0.83 | 13.57 | 0.45 | 16.93 | 0.77 | 12.26 |
| E2Former-V1 | 0.67 | 8.90 | 0.49 | 7.10 | 0.43 | 4.01 | 0.51 | 5.63 | 1.10 | 19.20 | 0.96 | 13.50 | 0.65 | 10.20 | 0.60 | 7.46 |
| eSEN-Direct | - | - | - | - | - | - | - | - | - | - | - | - | - | - | 0.56 | 10.98 |
| eSEN-Cons. | - | - | - | - | - | - | - | - | - | - | - | - | - | - | **0.23** | **6.36** |
| E2V2-Direct | 0.65 | 9.49 | 0.46 | 4.59 | 0.28 | 2.40 | 0.33 | 5.63 | 0.56 | 13.37 | 0.70 | 7.43 | 0.63 | 11.69 | 0.53 | 8.27 |
| E2V2-Cons. | **0.37** | **8.00** | **0.19** | **3.25** | 0.18 | 2.36 | 0.20 | 4.45 | 0.30 | 9.51 | 0.30 | 5.26 | 0.25 | 9.43 | 0.31 | 7.05 |

*Table 2.* **OMol25 Performance.** Val-Comp evaluation of Energy ($E$, meV/atom) and Force ($F$, meV/Å) prediction errors across domains and total. **Bold** and underline denote the best and second-best performers. Shaded rows indicate our proposed E2Former-V2 variants.

| DATASET | MODEL | VAL-COMP | | | | | | | | | |
|---|---|---|---|---|---|---|---|---|---|---|---|
| | | BIOMOLECULES | | ELECTROLYTES | | METALS | | ORGANICS | | TOTAL | |
| | | $E$ | $F$ | $E$ | $F$ | $E$ | $F$ | $E$ | $F$ | $E$ | $F$ |
| | eSEN-sm-d | 0.67 | 6.30 | 1.24 | 9.41 | 2.53 | 33.08 | 1.23 | 13.84 | 1.49 | 9.92 |
| | eSEN-sm-cons | 0.59 | **4.61** | **1.01** | **8.08** | 2.30 | 28.86 | **0.84** | **11.11** | 1.27 | **8.25** |
| All | UMA-S | **0.53** | 5.69 | 2.69 | 11.65 | 4.63 | 37.85 | 1.00 | 13.15 | 3.62 | 12.02 |
| | MACE-OMOL | 1.76 | 6.05 | 2.15 | 10.87 | **2.15** | 24.51 | 2.31 | 23.70 | 1.84 | 10.10 |
| | E2V2-Direct | 0.70 | 6.30 | 1.58 | 10.12 | 2.62 | 26.46 | 1.29 | 20.71 | 1.24 | 9.55 |
| | E2V2-Cons. | 1.01 | 4.96 | 1.52 | 10.11 | 2.18 | **24.40** | 1.26 | 19.69 | **1.16** | 8.38 |

*Table 3.* Forward runtime (ms) for the tensor product ($128 \times 0e + 128 \times 1e + 128 \times 2e) \otimes (1 \times 2e)$ on an H20 GPU, as a function of the number of tensor-product operations. EAAS is consistently the fastest across this range.

| Method \ #TP | 1k | 2k | 4k | 8k | 16k | 32k |
|---|---|---|---|---|---|---|
| e3nn (SO(3)) | 0.88 | 0.90 | 1.06 | 1.51 | 2.39 | 4.25 |
| cuEquivariance | 1.38 | 1.41 | 1.40 | 1.44 | 1.93 | 3.06 |
| OpenEquivariance | 0.55 | 0.68 | 1.12 | 1.86 | 3.31 | 6.20 |
| EAAS SO(2) (ours) | **0.15** | **0.22** | **0.35** | **0.65** | **1.20** | **2.25** |

*Table 4.* **End-to-end component ablation** of EAAS and the fused Triton kernel on a conservative model (single H20, average neighbor count $\sim$50). The Triton kernel provides the dominant speedup and is essential to avoid OOM; EAAS further improves throughput on top of it.

| Atoms $N$ | EAAS | Triton kernel | Samples/s |
|---|---|---|---|
| 5,000 | ✗ | ✗ | 1.40 |
| 5,000 | ✗ | ✓ | 3.31 |
| 5,000 | ✓ | ✗ | 1.55 |
| 5,000 | ✓ | ✓ | **4.24** |
| 10,000 | ✗ | ✗ | OOM |
| 10,000 | ✗ | ✓ | 1.65 |
| 10,000 | ✓ | ✗ | OOM |
| 10,000 | ✓ | ✓ | **2.12** |

gro and EquiformerV2, outperforming GotenNet by $\sim$4.5$\times$.

### 5.4. Molecular dynamics simulation.

We evaluated E2Former-V2 in molecular dynamics (MD) simulations using a custom-modified version of GROMACS 2025.3 (Abraham et al., 2015). All simulations were performed in the NVT ensemble at 300 K, and results were compared against MACE-OFF (Kovács et al., 2025).

For condensed-phase validation, we simulated periodic bulk water (216-H2O) and computed the oxygen-oxygen radial distribution function (RDF). As shown in Fig. 4a, E2Former-V2 closely matches the experimental RDF and shows improved agreement over MACE-OFF, indicating accurate

modeling of liquid water structure.

We further simulated a solvated protein system ($\sim$30k atoms) initialized from an AlphaFold-predicted(Abramson et al., 2024) structure for 2 ns. Figure 4b shows the TM-score relative to the experimental native structure, which remains consistently high, demonstrating stable dynamics and improved structural alignment in a realistic aqueous environment.

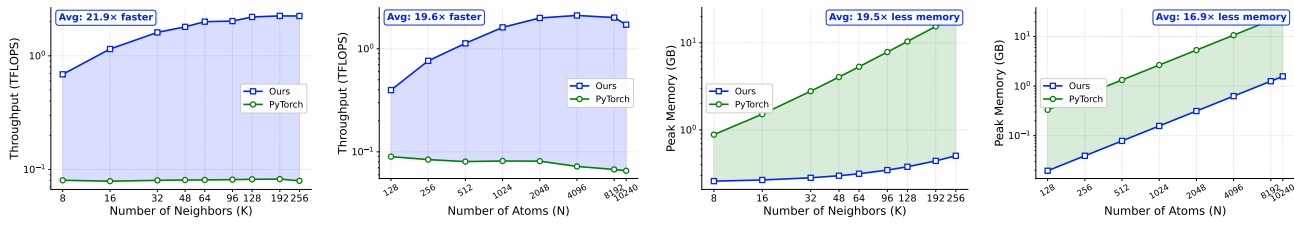

*(a)* TFLOPS vs. Neighbors    *(b)* TFLOPS vs. Atoms ($N$)    *(c)* Peak Memory vs. Neighbors    *(d)* Peak Memory vs. Atoms ($N$)

*Figure 3.* **Performance benchmarks of our on-the-fly equivariant attention kernels on H20 GPU. (a)(b)** Computational throughput (TFLOPS) as a function of the number of neighbors $K$ and atoms $N$, respectively. **(c)(d)** Peak GPU memory usage (GB) as a function of the number of neighbors $K$ and atoms $N$, respectively.

*Table 5.* **Inference Throughput Scaling.** Throughput (steps/s) is measured on a single H20 GPU. We compare methods in two categories: *Conservative* (forces via energy gradients, $F = -\nabla E$) and *Direct* (forces via a dedicated force head). **Bold** and underline denote the best and second-best performers within each category. ☐ Shaded ☐ columns indicate our proposed E2Former-V2 variants.

| ATOMS $N$ | CONSERVATIVE | | | | | | DIRECT | | | | |
|---|---|---|---|---|---|---|---|---|---|---|---|
| | eSCN | MACE OMOL | MACE-L | E2Former V1 | UMA-S | **E2Former V2** | GotenNet | EscAIP S | Allegro | Equiformer V2 | **E2Former V2** |
| **1,000** | 0.71 | 3.00 | 8.33 | 5.00 | 6.67 | **21.17** | 14.25 | 16.00 | 4.14 | 6.69 | **58.33** |
| **10,000** | OOM | OOM | OOM | 0.53 | 0.67 | **2.12** | 1.42 | 1.60 | OOM | OOM | **5.83** |
| **50,000** | OOM | OOM | OOM | OOM | 0.08 | **0.25** | 0.21 | 0.23 | OOM | OOM | **1.09** |
| **100,000** | OOM | OOM | OOM | OOM | 0.04 | **0.12** | 0.11 | OOM | OOM | OOM | **0.52** |

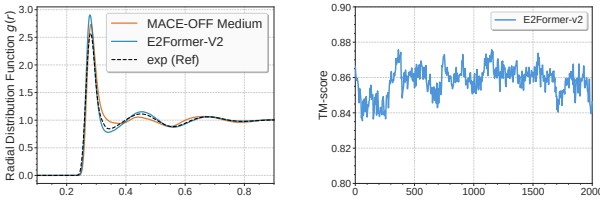

*(a)* Water Oxygen-Oxygen RDF    *(b)* Solvated protein TM-score

*Figure 4.* **Validation of E2Former-V2 in MD simulations.** (a) Oxygen–oxygen RDF from periodic bulk water simulations, compared with experimental reference data and MACE-OFF. (b) TM-score over time for a 2 ns MD simulation of a solvated protein system (∼30k atoms) initialized from an AlphaFold-predicted structure, showing stable dynamics and improved alignment with the experimental native conformation.

## 6. Conclusion

In this paper, we identify that the scalability of equivariant architectures is hindered by the memory bottlenecks of edge-centric tensor materialization. Then, we propose E2Former-V2, which integrates **Equivariant Axis-Aligned Sparsification (EAAS)** with a novel **On-the-Fly Equivariant Attention kernel**. This hardware-aware design strictly enforces node-centric computation and eliminates explicit edge tensors, thereby achieving linear activation memory. Experiments demonstrate that our method accelerates inference by **20x** while maintaining superior performance on molecular benchmarks.

## Impact Statement

Our work focuses on improving the efficiency and scalability of equivariant neural networks for molecular dynamics and property prediction. Potential societal impacts include accelerating the development of new pharmaceuticals and sustainable materials. We do not foresee any specific negative ethical consequences beyond the general dual-use risks associated with molecular design technologies.

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

## A. Latency Measurement Details

Figure 1 reports end-to-end forward latency of the same sparse attention pipeline (QK score computation + softmax over neighbors + value aggregation) under a fixed neighbor budget $K$ while sweeping the number of atoms $N$. Unless otherwise noted, the benchmark uses fp32 on a single GPU, with $K = 64$, $H = 16$, and $N \in \{128, 512, 2048, 8192, 32768\}$.

**Curve definitions.** **Traditional EGNNs** corresponds to a PyTorch edge-centric sparse implementation that explicitly gathers neighbor keys/values, materializing intermediate tensors of shape $N \times K \times H \times D$ (keys) and $N \times K \times H \times C$ (values), followed by QK reduction, softmax over the neighbor dimension, and a weighted value reduction.

**FlashAttention** corresponds to a sparse masked-attention baseline implemented using PyTorch `scaled_dot_product_attention` with a pre-built attention mask encoding the same $K$-neighbor pattern. The mask is constructed once per $N$ and reused across timing iterations so that mask construction is not included in the reported latency.

**Ours** corresponds to the proposed fused sparse implementation, where QK and value aggregation are computed using custom kernels, and the reduction over neighbors is performed on the fly without explicitly materializing edge-level intermediate tensors.

**Timing protocol.** For each $N$, we generate random inputs $(q, k, h')$ and neighbor indices $\mathbf{I} \in \mathbb{Z}^{N \times K}$. We run 10 warmup iterations followed by 50 timed iterations. Each iteration synchronizes the GPU (`torch.cuda.synchronize()`) before reading the wall-clock time. The plotted latency is the mean over timed iterations.

**Reference implementation snippets.** The following code sketch summarizes the three implementations used to generate Figure 1.

```
# Traditional EGNNs (PyTorch sparse): explicit gather + materialization
gk = key[idx].view(N, K, H, D)          # materialize N x K x H x D
scores = (query[:,None] * gk).sum(-1)   # N x K x H
alpha = softmax(scores, dim=1)          # N x K x H
gv = value[idx].view(N, K, H, C)        # materialize N x K x H x C
out = (alpha[...,None] * gv).sum(1)     # N x H x C

# FlashAttention (sparse): SDPA with pre-built K-neighbor mask
mask = build_mask_from_idx(idx)         # built once, not timed
out = scaled_dot_product_attention(q, k, v, attn_mask=mask)

# Ours: fused sparse kernels (QK + V) with on-the-fly reduction
scores = triton_sparse_qk(query, key, idx, gate, scale)    # N x K x H
alpha  = softmax(scores, dim=1)
out    = triton_sparse_v(value, alpha, idx)                # N x H x C
```

## B. Proof of Lemma 4.1 (Pole Sparsity)

In this subsection, we demonstrate that the geometric encodings vanish for all non-zero orders $m$ when expressed in the axis-aligned frame.

Recall the definition of the geometric encoding provided in Eq. (2):

$$\mathcal{R}_m^{(\ell)}(\vec{r}) = \|\vec{r}\|^\ell Y_{\ell,m}\left(\frac{\vec{r}}{\|\vec{r}\|}\right),\qquad(19)$$

where $Y_{\ell,m}$ denote the real spherical harmonics acting on the unit vector $\hat{n} = \vec{r}/\|\vec{r}\|$.

Consider a rotation $R \in \mathrm{SO}(3)$ that aligns the global $z$-axis with the vector $\vec{r}$. In this aligned frame, the unit vector becomes the basis vector along the $z$-axis:

$$\frac{R\vec{r}}{\|R\vec{r}\|} = \hat{e}_z = (0, 0, 1)^\top.\qquad(20)$$

The value of the real spherical harmonics $Y_{\ell,m}(\hat{n})$ is determined by the associated Legendre polynomials $P_\ell^{|m|}(\xi)$, where $\xi$ is the projection of the unit vector onto the $z$-axis (i.e., $\xi = \hat{n} \cdot \hat{e}_z = \cos\theta$). Specifically:

$$Y_{\ell,m}(\hat{n}) \propto P_\ell^{|m|}(\hat{n} \cdot \hat{e}_z). \tag{21}$$

In the aligned frame, we evaluate this at the pole $\hat{n} = \hat{e}_z$, yielding the argument $\xi = 1$. A fundamental property of the associated Legendre polynomials is:

$$P_\ell^k(1) \propto \delta_{k,0}. \tag{22}$$

For any non-zero order $m \neq 0$ (implying $|m| \geq 1$), the polynomial term contains a factor $(1 - \xi^2)^{|m|/2}$, which vanishes at $\xi = 1$.

Consequently, substituting this back into the definition of the geometric encoding:

$$\mathcal{R}_m^{(\ell)}(R\vec{r}) = \|\vec{r}\|^\ell Y_{\ell,m}(\hat{e}_z) \propto \delta_{m,0}. \tag{23}$$

This confirms that in the local axis-aligned frame, all magnetic components vanish except for the zero-order term ($m = 0$).

## C. Derivation of the EAAS Re-indexing Operator

This appendix provides the explicit form and derivation of the re-indexing operator $\mathcal{P}$ introduced in Definition 4.2 and used in Proposition 4.3. All derivations are exact and preserve SO(3) equivariance.

### C.1. Deriving the EAAS Re-indexing Rule

This subsection explains how the sparse re-indexing rule used to define $\mathcal{P}$ arises from standard Clebsch–Gordan (CG) structure after (i) aligning $\vec{r}$ to the $z$-axis and (ii) expressing features in the real SO(3) basis adopted throughout this paper.

**Complex vs. real CG conventions.** Many selection rules are most conveniently stated in the complex (physics) spherical-harmonic basis. To make this explicit, we denote the CG coefficients in the complex basis by $\bar{C}_{(\ell_i,m_i),(\ell_f,m_f)}^{(\ell_o,m_o)}$, and the CG coefficients in our real SO(3) basis by $C_{(\ell_i,m_i),(\ell_f,m_f)}^{(\ell_o,m_o)}$. The two conventions are related by a fixed change-of-basis within each degree $\ell$ that mixes the pair of orders $\{m, -m\}$.

Concretely, let $z_m^{(\ell)}$ denote the complex basis and $x_m^{(\ell)}$ the real basis used in the paper. They are related (for each $\ell$) by

$$x_m^{(\ell)} = \begin{cases} \frac{i}{\sqrt{2}}\left(z_m^{(\ell)} - (-1)^m z_{-m}^{(\ell)}\right), & m < 0, \\ z_0^{(\ell)}, & m = 0, \\ \frac{1}{\sqrt{2}}\left(z_m^{(\ell)} + (-1)^m z_{-m}^{(\ell)}\right), & m > 0. \end{cases} \tag{24}$$

This change-of-basis is fixed (input-independent) and depends only on $(\ell, m)$.

**Alignment implies $m_f = 0$.** Let $R \in \text{SO}(3)$ align the global $z$-axis with $\vec{r}$. By Lemma 4.1, in the aligned frame the geometric encoding has only the zero-order component:

$$\mathcal{R}_m^{(\ell_f)}(R\vec{r}) \propto \delta_{m,0}, \qquad \text{i.e.,} \qquad m_f = 0. \tag{25}$$

Therefore, the CG contraction in the aligned frame always couples with $(\ell_f, 0)$.

**Selection rule in the complex basis.** In the complex basis, CG coefficients satisfy the standard order constraint:

$$\bar{C}_{(\ell_i,m_i),(\ell_f,m_f)}^{(\ell_o,m_o)} \neq 0 \quad \Rightarrow \quad m_o = m_i + m_f. \tag{26}$$

With $m_f = 0$, this implies that in the complex basis only terms with $m_o = m_i$ can contribute:

$$\bar{C}_{(\ell_i,m_i),(\ell_f,0)}^{(\ell_o,m_o)} \neq 0 \quad \Rightarrow \quad m_o = m_i. \tag{27}$$

Moreover, the complex CG coefficients obey the symmetry relation

$$\bar{C}^{(\ell_o,m_o)}_{(\ell_i,m_i),(\ell_f,m_f)} = (-1)^{\ell_i+\ell_f+\ell_o} \bar{C}^{(\ell_o,-m_o)}_{(\ell_i,-m_i),(\ell_f,-m_f)}. \tag{28}$$

Setting $m_f = 0$ and using $m_o = m_i$ yields, for any $m$,

$$\bar{C}^{(\ell_o,m)}_{(\ell_i,m),(\ell_f,0)} = (-1)^{L_\Sigma} \bar{C}^{(\ell_o,-m)}_{(\ell_i,-m),(\ell_f,0)}, \qquad L_\Sigma := \ell_i + \ell_f + \ell_o. \tag{29}$$

**Change-of-basis induces parity-dependent sparsity.** Equation (24) mixes the $\{m, -m\}$ pair when passing from the complex basis to the real basis. For any fixed $m \neq 0$, consider the $2 \times 2$ block of coefficients associated with the ordered pair $\{-m, m\}$ (input and output). Using Eq. (27)–(29) and applying the change-of-basis on both the input and output irreps yields a parity-dependent collapse:

- If $L_\Sigma$ is even, the real-basis coupling is diagonal in the pair $\{-m, m\}$, i.e., only the mapping $m_i = m_o$ survives.

- If $L_\Sigma$ is odd, the real-basis coupling becomes off-diagonal in the pair $\{-m, m\}$, i.e., only the mapping $m_i = -m_o$ survives, and the surviving entry acquires a fixed sign factor depending on $m_o$.

Under our convention, this fixed sign appears as the factor $-2(-1)^{m_o}$.

The case $m = 0$ is consistent with the above rule: Eq. (29) implies that the $m_o = 0$ component vanishes when $L_\Sigma$ is odd.

**Resulting re-indexing rule.** Combining the above observations, the action of $\mathcal{P}$ in the aligned frame reduces to a deterministic re-indexing (within each $\ell$-block) with no dense summation over magnetic indices:

$$\left(\mathcal{P}(\tilde{h})\right)^{(\ell_o)}_{m_o} = \left[ \begin{cases} C^{(\ell_o,m_o)}_{(\ell_i,m_i),(\ell_f,0)} \tilde{h}^{(\ell_i)}_{m_i}, & \text{if } L_\Sigma \text{ is even,} \\ -2(-1)^{m_o} C^{(\ell_o,m_o)}_{(\ell_i,-m_i),(\ell_f,0)} \tilde{h}^{(\ell_i)}_{-m_i}, & \text{if } L_\Sigma \text{ is odd,} \end{cases} \right], \tag{30}$$

where $L_\Sigma = \ell_i + \ell_f + \ell_o$, and the repeated index $m_i$ is implicitly summed over. This is exactly the re-indexing structure used in Definition 4.2 and Proposition 4.3.

### C.2. Proof of Proposition 4.3

We derive the aligned-frame form of $\mathcal{P}$ from the Clebsch–Gordan contraction underlying the SO(3)-equivariant tensor product.

**Commuting the rotation.** Consider the SO(3)-equivariant tensor product between node features $h^{(\ell_i)}$ and geometric encodings $\mathcal{R}^{(\ell_f)}(\vec{r})$. By equivariance, for any rotation $R \in \mathrm{SO}(3)$,

$$h \otimes \mathcal{R}(\vec{r}) = D_{R^{-1}}\Big(D_R h \ \otimes \ D_R \mathcal{R}(\vec{r})\Big). \tag{31}$$

Let $\tilde{h} = D_R h$ denote the node features expressed in the aligned frame.

**Pole sparsity.** Choosing $R$ such that the global $z$-axis is aligned with $\vec{r}$, Lemma 4.1 implies

$$\mathcal{R}^{(\ell_f)}_m(R\vec{r}) \propto \delta_{m,0}. \tag{32}$$

Thus, the geometric encoding contains only the zero-order component $m_f = 0$ in the aligned frame.

**Clebsch–Gordan expansion.** Projecting the tensor product in Eq. (31) to output degree $\ell_o$ and order $m_o$ yields

$$\left(h \otimes \mathcal{R}(\vec{r})\right)^{(\ell_o)}_{m_o} = \sum_{m_i=-\ell_i}^{\ell_i} \tilde{h}^{(\ell_i)}_{m_i} C^{(\ell_o,m_o)}_{(\ell_i,m_i),(\ell_f,0)}. \tag{33}$$

**Parity selection rule.** The Clebsch–Gordan coefficients in Eq. (33) obey a parity selection rule. Let $L_\Sigma = \ell_i + \ell_f + \ell_o$. If $L_\Sigma$ is even, the coefficient is non-zero only when $m_i = m_o$. If $L_\Sigma$ is odd, the coefficient is non-zero only when $m_i = -m_o$, up to a fixed sign convention, which in our notation appears as the factor $-2(-1)^{m_o}$ in Eq. (30). All other combinations vanish.

Substituting this rule into Eq. (33) recovers exactly the operator form in Eq. (30), completing the proof.

### C.3. Concrete Examples

We illustrate the re-indexing operator for low-degree cases.

$\ell_i = 1, \ell_f = 1, \ell_o = 0$. Here $L_\Sigma = 2$ is even, and the only output order is $m_o = 0$. The re-indexing rule yields

$$(\mathcal{P}(\tilde{h}))_0^{(0)} = C_{(1,0),(1,0)}^{(0,0)} \, \tilde{h}_0^{(1)}.$$

Thus, the scalar output depends only on the $m = 0$ component of the input.

$\ell_i = 1, \ell_f = 1, \ell_o = 1$. Here $L_\Sigma = 3$ is odd. The re-indexing rule maps each output order to the opposite input order:

$$(\mathcal{P}(\tilde{h}))_{m_o}^{(1)} = -2(-1)^{m_o} \, C_{(1,-m_o),(1,0)}^{(1,m_o)} \, \tilde{h}_{-m_o}^{(1)}.$$

The $m_o = 0$ component vanishes due to the Clebsch–Gordan coefficient, while the $m_o = \pm 1$ components are obtained by swapping the corresponding input orders.

These examples illustrate that $\mathcal{P}$ acts as a deterministic re-indexing with scaling, rather than a dense summation, within each representation block.

## D. Relation to EquiformerV2/eSEN/eSCN and the Role of Non-linearity

A natural concern when comparing our attention mechanism with EquiformerV2 (Liao et al., 2023) and eSEN (Fu et al., 2025a) is whether the non-linearity along the value path is removed. We clarify that the non-linearity is *not* removed, but rather *reorganized* into the attention construction itself.

**Our formulation.** Instead of applying a gated/$S^2$ non-linearity directly to edge-level value features, we first construct an edge feature

$$\mathbf{Fea}_{ij} = \sigma\left( (\mathbf{Q}_i \,\|\, \mathbf{V}_j)^{(\ell)} \odot \mathbf{r}_{ij}^{(\ell)} \right), \tag{34}$$

where $\|$ denotes channel-wise concatenation, $\odot$ is a channel-wise product within each angular order $\ell$, and $\sigma$ is a pointwise non-linearity. The attention coefficient is then computed as

$$\alpha_{ij} = \mathrm{softmax}_{j \in \mathcal{N}(i)}(\mathrm{vec2scalar}(\mathbf{Q}_i) \cdot \mathrm{vec2scalar}(\mathbf{K}_j) \cdot \mathbf{Fea}_{ij} + b(\|\mathbf{r}_{ij}\|)) \cdot \phi(\|\mathbf{r}_{ij}\|), \tag{35}$$

where $\mathrm{vec2scalar}(\cdot)$ denotes the SO(3)-invariant scalar reduction used in Eq. (6), and $b$ and $\phi$ are radial gating function. Equivalently, this corresponds to the `tp_type=QKdotS_alpha+triton` configuration reported in Table 10.

**Where the non-linearity lives.** In Eqs. (34)–(35) the non-linearity is injected through (i) $\sigma$ applied to the coupled feature $(Q\|V) \odot r$, and (ii) the radial basis function $\phi(r_{ij})$, which modulates the attention weight. The resulting $\alpha_{ij}$ therefore plays a role analogous to a non-linear *attention bias* rather than a value-path activation.

**Contrast with EquiformerV2/eSEN/eSCN.** EquiformerV2 and eSEN apply SO(2) conv $\rightarrow$ $S^2$ (or gated) activation $\rightarrow$ SO(2) conv along the edge value path. This introduces an explicit equivariant non-linearity directly on the value features, which contributes additional expressivity but also materializes edge-level intermediates and forces edge-centric computation. Our design trades an explicit edge-level value-path non-linearity for a more HBM-efficient attention formulation (Eqs. (34)–(35)), while still retaining non-linearity via $\sigma$ and the radial features. This choice is a deliberate efficiency/expressivity trade-off: we empirically observe comparable accuracy on SPICE and OMol25 (Tables 1 and 2), while substantially improving throughput and memory scaling.

Moreover, we compare the dominant cost terms of e EquiformerV2/eSEN/eSCN and E2Former-V2 under a common notation, where $h$ is the channel dimension, $L$ is the maximum degree, $N$ is the number of atoms, and $k$ is the average neighbor count.

**E2Former-V2 (node-centric, $O(N)$).** The tensor product $(h \times 0e + h \times 1e + \cdots + h \times Le) \otimes (1 \times L2e)$ decomposes into:

- Wigner-$D$ transform: $(L+1)^4 \cdot h$,
- EAAS re-indexing: $(L+1)^2 \cdot h$,
- linear mixing: $(L+1)^2 \cdot h \cdot h$.

When $h \gg (L+1)^2$, the linear term dominates; otherwise the Wigner-$D$ transform is the leading cost.

**EquiformerV2/eSEN/eSCN (edge-centric, $O(E) = O(kN)$).** The tensor product $(h \times 0e + \cdots + h \times Le) \otimes (1 \times 0e + \cdots + 1 \times Le)$ incurs:

- Wigner-$D$ transform: $(L+1)^4 \cdot h$,
- SO(2) convolution (fully connected coupling): $(L+1)^2 \cdot L \cdot h \cdot h$.

In practice, when $h \gg L$, the SO(2) convolution dominates the edge-level cost; otherwise the Wigner-$D$ term dominates. Crucially, these edge-level costs scale with $|\mathcal{E}| = kN$ in eSCN, whereas the corresponding terms in E2Former-V2 scale with $|\mathcal{V}| = N$ due to our node-centric formulation.

## E. Neighbor-Density Sensitivity

A natural follow-up question is whether the benefit of E2Former-V2's node-centric design—converting edge complexity into node complexity—also materializes when the graph is very sparse (e.g. $k = 2$ or $5$ neighbors). In E2Former-V2, the node-level cost is $\mathcal{O}(N)(L+1)^2 h^2$, while edge-level terms include attention $\mathcal{O}(kN)h$, bias $\mathcal{O}(kN)h^2$, and aggregation $\mathcal{O}(kN)(L+1)^2 h$. For small $k$, the bottleneck is dominated by the node-level term; as $k$ grows, edge-level terms become significant. Edge-centric baselines, in contrast, always carry an edge-level linear/nonlinear cost of at least $\mathcal{O}(kN)(L+1)^2 h^2$, so their runtime depends much more strongly on $k$.

Tables 6 and 7 confirm this prediction on a 10,000-atom system.

| Neighbors $k$ | Runtime (ms/step) | Samples/s |
|:---:|:---:|:---:|
| 2 | 85.4 | 11.7 |
| 5 | 89.4 | 11.2 |
| 10 | 96.1 | 10.4 |
| 20 | 118.3 | 8.4 |
| 50 | 184.9 | 5.4 |
| 100 | 282.0 | 3.5 |

*Table 6.* E2Former-V2 inference speed as a function of the average neighbor count $k$, on a 10,000-atom system.

| Neighbors $k$ | Runtime (ms/step) | Samples/s |
|:---:|:---:|:---:|
| 2 | 70.4 | 14.2 |
| 5 | 121.9 | 8.2 |
| 10 | 212.2 | 4.7 |
| 20 | 398.1 | 2.5 |
| 50 | 962.9 | 1.0 |
| 100 | 1872.7 | 0.5 |

*Table 7.* Representative edge-centric method on the same 10,000-atom system. Runtime scales nearly linearly with $k$.

When $k$ is very small (e.g. $k = 2$), the edge-centric baseline can even be slightly faster than E2Former-V2, since its workload is tiny and node-level overhead dominates E2Former-V2. However, the gap grows rapidly once $k \gtrsim 10$. In practical MLIP settings, $k$ is typically $\sim 50$–$100$ for liquid water and $\sim 30$–$50$ for OC20/OC22 under a 5–6 Å cutoff, which is precisely the regime where E2Former-V2 offers the largest speedup.

## F. Datasets

This appendix summarizes the two datasets used in our evaluation: SPICE and OMol25.

**SPICE (Small-molecule/Protein Interaction Chemical Energies).** SPICE focuses on quantum-mechanical energetics relevant to medicinal chemistry settings, particularly small molecules in protein-like environments and related non-covalent interactions. The dataset contains over 1.1M conformations spanning drug-like small molecules, dimers, dipeptides, and solvated amino acids, covering both neutral and charged species and multiple interaction motifs. For each conformation, SPICE provides high-quality quantum chemical labels including energies and forces (and additional molecular properties such as multipole moments and bond orders), computed at the $\omega$B97M-D3(BJ)/def2-TZVPPD level of theory.

**OMol25 (Open Molecules 2025).** OMol25 is a large-scale, high-accuracy quantum chemistry dataset designed for training and evaluating foundation-scale molecular models across broad chemical space. It contains more than 100M DFT single-point calculations at the $\omega$B97M-V/def2-TZVPD level of theory, comprising roughly 83M unique molecular systems. OMol25 provides exceptional chemical and elemental diversity (including a wide range of intra- and intermolecular interactions, conformers, variable charge/spin states, and reactive structures), and includes systems up to approximately 350 atoms.

## G. OC20 S2EF-2M Results

To further assess accuracy on widely benchmarked catalysis data, we train E2Former-V2 on the OC20 S2EF-2M subset and compare against commonly reported baselines (Table 8). E2Former-V2 achieves energy and force accuracy comparable to strong equivariant baselines (e.g. eSCN, EquiformerV2), while retaining the throughput advantages reported in the main paper. This confirms that the efficiency-focused design of E2Former-V2 does not come at the cost of accuracy on standard catalysis benchmarks.

| Model | # Params (M) | Energy MAE (meV) | Force MAE (meV/Å) |
|---|---|---|---|
| GemNet-dT | 31 | 358 | 29.50 |
| GemNet-OC | 38 | 286 | 25.70 |
| SCN | 126 | 279 | 21.90 |
| eSCN | 51 | 283 | 20.50 |
| EquiformerV2 | 85 | 285 | 20.46 |
| E2Former | 33 | **275** | 21.90 |
| E2Former-V2 | 54 | 287 | 21.80 |

*Table 8.* OC20 S2EF-2M validation results. E2Former-V2 achieves accuracy comparable to mainstream equivariant baselines while providing substantially higher throughput at larger system sizes (see main-paper Table 5).

## H. Inference throughput benchmark setup

This section describes the experimental setup used to produce Table 5.

**Hardware and metric.** All throughput numbers are measured on a single NVIDIA H20 GPU. We report *throughput* as steps per second (QPS), computed as $\mathrm{QPS} = \mathtt{steps/time}$, where $\mathtt{time}$ is the wall-clock duration of a fixed number of forward passes.

**System generation.** Following the UMA benchmark methodology, synthetic molecular systems are generated using an FCC carbon crystal from ASE. For a target atom count $N$, we construct an FCC carbon supercell and uniformly sample $N$ atoms without replacement. We use a lattice constant $a = 3.8$, which yields approximately $\sim 50$ neighbors per atom under a 6 Å cutoff in the original UMA setting. We evaluate system sizes $N \in \{1\mathrm{k}, 10\mathrm{k}, 50\mathrm{k}, 100\mathrm{k}\}$. Inputs are converted to the model format using the same batching pipeline (`collate_fn`) as in training.

**Model invocation and force modes.** We benchmark inference in two categories: *Conservative* models compute forces via energy gradients, $F = -\nabla E$, which requires autograd through the energy head. *Direct* models predict forces with a

dedicated force head. For the Direct category, we disable autograd-force computation (`AutoGradForce=False`) and report the runtime of direct force prediction. All runs use `model.eval()` and `torch.no_grad()`.

**Timing protocol.** For each system size $N$, we run 10 warmup iterations followed by 10 timed iterations. Timing uses `timeit.timeit` over the complete forward call, with `torch.cuda.synchronize()` inside the timed function to ensure accurate GPU measurement. We report the average time per step and QPS.

**Memory reporting and OOM handling.** We reset peak CUDA memory statistics before benchmarking each $N$ (`torch.cuda.reset_peak_memory_stats()`) and report the peak allocated memory when available. If an evaluation fails due to out-of-memory (OOM) or other runtime errors, the corresponding entry is marked as OOM in Table 5.

## I. Training Configuration

This section summarizes the training configurations used for E2Former-V2. We consider two force modes: *Direct* predicts forces with a dedicated force head (`AutoGradForce=False`), and *Conservative* computes forces via energy gradients (`AutoGradForce=True`). Details are given in Table 9 and Table 10.

**Training schedule and initialization.** **Direct** is trained for **17 epochs** on **OMol25** and then **100 epochs** on **SPICE**. **Conservative** is initialized from the **Direct checkpoint** and continues training with the same schedule (**17 epochs** on **OMol25**, then **100 epochs** on **SPICE**). For Conservative, we load the Direct checkpoint via `ckpt_path`.

| Hyperparameter | Direct | Conservative |
|---|---|---|
| Force mode | `AutoGradForce=False` | `AutoGradForce=True` |
| Loss | `loss_fn=atoml2mae` | `loss_fn=atoml2mae` |
| Learning rate | $8 \times 10^{-4}$ | $6 \times 10^{-4}$ |
| Weight decay | $1 \times 10^{-3}$ | $1 \times 10^{-3}$ |
| Energy loss weight | 4 | 3 |
| Force loss weight | 30 | 10 |
| Gradient clipping | `clip_grad_norm=100` | `clip_grad_norm=20` |
| Micro-batch size | `atom:2048` | `atom:2048` |
| Scheduler | `groupWarmupDecayLR` | `groupWarmupDecayLR` |
| Warmup steps | 8,000 | 8,000 |
| Parallel strategy | `ddp` | `ddp` |

*Table 9.* Training hyperparameters for E2Former-V2 under Direct and Conservative force modes.

| Model / Backbone setting | Value |
|---|---|
| Backbone | `e2former` |
| # layers | `backbone_config.num_layers=4` |
| Clustering | `backbone_config.with_cluster=False` |
| Node embedding irreps | `"256x0e+256x1e+256x2e"` |
| Head irreps | `"2x0e+2x1e+2x2e"` |
| Scalar head dim | `backbone_config.attn_scalar_head=8` |
| # attention heads | `backbone_config.num_attn_heads=128` |
| Radial basis | `backbone_config.number_of_basis=256` |
| Cutoff radius (Å) | `backbone_config.max_radius=6` |
| Max neighbors | `backbone_config.max_neighbors=32` |
| Attention dropout | `backbone_config.alpha_drop=0` |
| Basis type | `gaussiansmear` |
| Normalization | `rms_norm_sh` |
| Attention type | `so2-first-order` |
| Tensor product impl. | `tp_type=QKdotS_alpha+triton` |
| Edge embedding | `default` |
| FFN type | `s3` |
| Encoder | `default` |

*Table 10.* E2Former-V2 backbone hyperparameters (shared by both Direct and Conservative runs).

## J. Comparison with Dense FlashAttention and the Role of Triton Fusion

We further clarify the role of the fused Triton kernel and compare our sparse equivariant attention against dense FlashAttention-style implementations.

**Memory versus compute regimes.** With $E \approx kN$ edges, the arithmetic complexity of sparse attention remains $\mathcal{O}(Nk)$ regardless of implementation. However, a naive sparse implementation explicitly materializes edge-level $Q/K/V$ tensors in HBM, while the fused Triton implementation recomputes them on the fly inside registers/SRAM. Table 11 summarizes the difference.

| Aspect | Sparse (no Triton) | Sparse + Triton | SDPA-Dense |
|---|---|---|---|
| Attention type | Sparse | Sparse | Dense |
| Compute complexity | $\mathcal{O}(Nk)$ | $\mathcal{O}(Nk)$ | $\mathcal{O}(N^2)$ |
| Memory complexity | $\mathcal{O}(Nk)$ | $\mathcal{O}(N)$ | $\mathcal{O}(N)$ |
| Edge activations | Materialized per edge (HBM) | Recomputed on-the-fly | Recomputed on-the-fly |
| Execution regime | Memory-bound | Compute-bound | Compute-bound |
| Practical scaling | OOM beyond ∼5k atoms | Scales to ∼100k atoms | Quadratic scaling |

*Table 11.* Comparison between naive sparse attention, our fused sparse Triton kernel, and dense FlashAttention-style SDPA. The Triton kernel preserves the $\mathcal{O}(Nk)$ arithmetic complexity while eliminating edge-level HBM materialization, reducing memory usage from $\mathcal{O}(Nk)$ to $\mathcal{O}(N)$.

**Why fusion matters empirically.** In a vanilla PyTorch sparse implementation, $Q$, $K$, and $V$ must be explicitly gathered for every edge, producing intermediate tensors of shape $N \times K \times H \times D$. This quickly becomes memory-bound and causes OOM beyond ∼5k atoms on our hardware. In contrast, the fused Triton kernel recomputes $Q/K/V$ on the fly per neighbor block without materializing edge-level tensors, allowing scaling to ∼100k atoms (Tables 4 and 5).

**Comparison with dense FlashAttention.** To contextualize the absolute throughput numbers of our sparse equivariant attention kernel, we additionally benchmark dense FlashAttention-style implementations using PyTorch `scaled_dot_product_attention` (SDPA) on the same NVIDIA H20 GPU. We compare: (i) our fused sparse Triton kernel, (ii) SDPA with a $K$-neighbor mask (dense kernel with sparse masking), and (iii) fully dense SDPA. Table 12 reports per-kernel throughput.

| $N$ | head | $D$ | Sparse + Triton | SDPA-Mask | SDPA-Dense |
|---|---|---|---|---|---|
| 128 | 16 | 8 | 0.015 | 0.051 | 0.191 |
| 512 | 16 | 8 | 0.066 | 0.117 | 1.670 |
| 2048 | 16 | 8 | 0.235 | 0.088 | 4.420 |
| 8192 | 16 | 8 | 0.506 | 0.029 | 6.250 |
| 32768 | 16 | 8 | 0.609 | 0.007 | 6.470 |
| 128 | 16 | 64 | 0.110 | 0.345 | 1.330 |
| 512 | 16 | 64 | 0.433 | 0.792 | 11.100 |
| 2048 | 16 | 64 | 1.371 | 0.535 | 26.900 |
| 8192 | 16 | 64 | 1.995 | 0.191 | 37.300 |
| 32768 | 16 | 64 | 1.951 | 0.051 | 38.600 |

*Table 12.* Throughput (TFLOPS) of our fused sparse equivariant attention kernel, PyTorch SDPA with neighbor masking (SDPA-Mask), and fully dense SDPA (SDPA-Dense) on an NVIDIA H20 GPU. $N$ denotes the number of atoms (sequence length), and the feature dimension is head $\times D$.

**Discussion.** When $D = 8$ (feature dimension 128), the dominant attention GEMMs ($QK^\top$ and $AV$) are extremely skinny, leading to poor Tensor Core utilization even in dense attention. As $D$ increases to 64, the tile sizes become substantially more favorable for Tensor Core execution, enabling better blocking, vectorization, and SM occupancy, and thus significantly higher throughput.

Dense FlashAttention remains substantially faster because it operates on fully regular dense tiles and benefits from highly optimized Tensor Core pipelines. In contrast, our sparse Triton kernel performs irregular neighbor gathers and scatter-style

reductions, which are fundamentally more difficult to optimize. Moreover, our implementation currently operates in FP32 (H20 peak ∼44 TFLOPS), whereas dense FlashAttention heavily benefits from lower-precision execution.

Nevertheless, the sparse formulation provides dramatically better asymptotic memory scaling for atomistic systems. Dense SDPA scales quadratically with $N$ and becomes impractical for large molecular systems, whereas our fused sparse kernel maintains linear scaling with the neighbor count $k$ and can scale to ∼100k atoms in practice.

## K. detailed analysis of MD simulation

To validate physical fidelity in condensed-phase systems, we first considered bulk water simulations under periodic boundary conditions. A simulation box containing 216 water molecules was evolved, and the oxygen–oxygen radial distribution function (RDF) was computed and compared against experimental reference data. As shown in Fig. 4a, E2Former-V2 closely reproduces the experimental RDF, yielding improved agreement over the MACE-OFF baseline, particularly in the first solvation shell. This result demonstrates that E2Former-V2 accurately captures many-body interactions and hydrogen-bonding structures in liquid water over long-time dynamics.

We further simulated a solvated protein system (∼30k atoms) initialized from an AlphaFold-predicted structure and evolved for 2 ns in explicit water. Figure 4b reports the time evolution of the TM-score with respect to the experimentally resolved native structure. The consistently high TM-score throughout the trajectory indicates that E2Former-V2 maintains structural stability and guides the system toward a conformation more closely aligned with the experimental reference, demonstrating its robustness for large-scale biomolecular simulations.

