# OpenReview forum: "E2Former-V2: On-the-Fly Equivariant Attention  with Linear Activation Memory"
_ICML.cc/2026/Conference — ICML 2026 regular_

### Official Review · Reviewer_bP3d · 2026-03-12

**Soundness:** 2
**Presentation:** 2
**Significance:** 2
**Originality:** 2
**Overall Recommendation:** 3
**Confidence:** 4

**Summary:**

This paper proposes E2Former-V2 model, which uses (1) Equivariant Axis-Aligned Sparsification to simplify tensor products and (2) Triton kernels to optimize the computation of the attention in the model.

**Compliance With Llm Reviewing Policy:**

Affirmed.

**Final Justification:**

The rebuttal confirmed most of my concerns regarding what the contributions are made in this paper.
I appreciate the authors' acknowledgement and would encourage the authors to make corresponding updates to the paper to better reflect the actual contributions.

In brief, the contributions are having a better Triton implementation for the simplified attention. The speed is due to the kernel while the degradation in accuracy is because of the model.
This is not a genuine weakness as implementation-level optimization is orthogonal to model design and can be helpful for delpoying powerful (yet slow) models.

However, my biggest concern is the transparency of the paper and the original presentation. The authors do not seem to be very clear about the contributions and limitations.
Given this, I lean toward remaining my original score.

**Key Questions For Authors:**

1. What is the end-to-end speedup after using the Triton kernel?
2. What is the key difference from the attention in EquiformerV2? Seems that the non-linear function in value vectors is removed, resulting in degraded performance compared to models like eSEN.
3. What are the trade-offs of the efficient attention? I think similar Triton kernels can be applied to other equivariant networks (similar to fusing the graph convolution in OpenEquivariance (earlier work))
4. Results on other well-benchmarked datasets like OC20 S2EF-2M?

**Limitations:**

Please see Questions

**Strengths And Weaknesses:**

Strength
1. The speedup of Triton kernels seems to be great.


---

Weakness
1. It would be better to use "efficiency" instead of "scalability" in the abstract.
2. Introduction on page 1: the explanation of N^2 seems not to be consistent with Figure 1 -- FlashAttention has latency grow faster than EGNNs.
3. Section 4.2 seems to be the reduced version of the attention in Equiformer while there is no reference.
3. Lemma 4.1 seems to be the same as what eSCN proposed while it does not reference eSCN.
4. The reference of eSEN in Section 5.1 on Page 6 is wrong.
5. The representation of Figure 3 can be improved, and similar results are already in the work of eSCN.
6. Table 3 should uses the original name of different works rather than abbreviations.
7. Empirical results are not strong.
8. The idea of customized kernels is also in cuEquivariance and OpenEquivariance while the work does not mention them and compare to them.
9. Detailed architecture and the meaning of values in Table 5.
10. Overall the presentation of the paper can be greatly improved.
11. Hard to tell the novelty / contribution of the work except the Triton kernel to speed up attention.

---

> ### Author Rebuttal · Authors · 2026-03-30
>
> Thank you for your detailed and helpful feedback. We address your key concerns below and outline the improvements we will make in the revision.
>
> **W1:** Thank you for the suggestion. We will clarify the distinction between efficiency (kernel/runtime) and scalability (large-scale behavior) in the revision..
>
> **W2:** The “FlashAttention” curve uses PyTorch scaled_dot_product_attention with a K-neighbor mask, so its theoretical complexity is O(kN). In practice, latency can grow faster due to masking overhead.
>
> **W3:** Please refer to Q2 for more details (due to space limits).
>
> **W4:** We will add this reference.
>
> **W5:** Thank you for catching this, and we will correct it in the revision.
>
> **W6:** Thank you for the suggestion. We will improve Figure 3. However, this is not a repetition of eSCN, as E2Former-V2 and eSCN target different regimes.
>
> E2Former-V2 uses  (h × 0e + h × 1e + ... + h × l e) ⊗ (1 × l 2e),  and is an atom-level O(N) method, with cost dominated by linear terms N(l+1)^2×h^2.
>
> By contrast, eSCN uses  (h × 0e + h × 1e + ... + h × l e) ⊗ (1 × 0e + ... + 1 × l e),  and is an edge-level O(E) method, dominated by SO(2) coupling E(l+1)^2×l×h^2.
>
> **W7:** We will replace abbreviations in Table 3 with original method names.
>
> **W8:** We appreciate this concern. Our goal is to improve the practical scalability of equivariant models while maintaining competitive accuracy. On SPICE and OMol25, E2Former-V2 achieves comparable accuracy. On efficiency, it is about 3× faster than the fastest prior network.
>
> **W9:** We conducted additional comparisons. For the regime most relevant to our applications, i.e., feature = (128x0e + 128x1e + 128x2e) with SH = (1x1e)/(1x2e), SO2-EAAS is the fastest, followed by cuEquivariance, e3nn, and OpenEquivariance.
>
> **Table 1. (128x0e + 128x1e + 128x2e) ⊗ (1x2e), runtime (ms)(1e results omitted due to space)**
>
> | method\number of⊗ | 1000 | 2000 | 4000 | 8000 | 16000 | 32000 |
> | --- | --- | --- | --- | --- | --- | --- |
> | so2 | 0.15 | 0.22 | 0.35 | 0.65 | 1.20 | 2.25 |
> | so3 e3nn | 0.88 | 0.90 | 1.06 | 1.51 | 2.39 | 4.25 |
> | so3 cuequivariance | 1.38 | 1.41 | 1.40 | 1.44 | 1.93 | 3.06 |
> | so3 openequivariance | 0.55 | 0.68 | 1.12 | 1.86 | 3.31 | 6.20 |
>
> **W10:** Thank you for the suggestion. We will define the quantities in Table 5 in the revision.
>
> **W12:** Thank you for the comment. The novelty of this work is not merely “using Triton to speed up attention,” but showing that EAAS + fused kernel optimization + scalable end-to-end design make equivariant MLFFs practical for much larger molecular systems. Our key contributions are:
>
> 1. EAAS + Triton fused kernel: together they reduce the cost of equivariant computation and remove the main edge-level runtime/memory bottleneck.
> 2. End-to-end scalability: this leads to about 3× higher throughput than the fastest prior network on realistic large-scale workloads.
> 3. Practical impact: this enables large-scale MD applications, including large water-box simulations and, to our knowledge, the first stable MD refinement of AlphaFold-predicted protein structures in this class of equivariant MLFFs (up to 30k atoms).
>
> **Q1:** We performed an end-to-end ablation on an H20 machine (avg. neighbor ~50), comparing with/without EAAS and the Triton fused kernel:
>
> | Atom Count | EAAS | Triton | Samples/s |
> | --- | --- | --- | --- |
> | 5000 | ✗ | ✗ | 1.40 |
> | 5000 | ✗ | ✓ | 3.31 |
> | 5000 | ✓ | ✗ | 1.55 |
> | 5000 | ✓ | ✓ | 4.24 |
> | 10000 | ✗ | ✗ | OOM |
> | 10000 | ✗ | ✓ | 1.65 |
> | 10000 | ✓ | ✗ | OOM |
> | 10000 | ✓ | ✓ | 2.12 |
>
> The Triton fused kernel provides the main gain by removing the edge-level bottleneck, while EAAS brings additional improvement. The full system achieves the best throughput. For larger systems, see Table 3.
>
> **Q2:** Our attention differs from EquiformerV2 in two ways. First, Equiformer-style attention builds scores from tensor products between node-pair irreps and r_{ij}, while ours builds scalar queries/keys from invariant summaries across angular orders and applies dot-product attention. Second, we do not remove non-linearity; it is still introduced through the attention construction and radial basis functions similar like eSEN, and EquiformerV2. Empirically, we achieve comparable accuracy on SPICE and OMol25.
>
> **Q3:** There is no accuracy trade-off in our attention design: the speedup comes from an HBM-efficient implementation that reduces memory movement without changing the computation. Similar Triton kernels could also be applied to other equivariant graph networks.
>
> **Q4:** We added experiments on OC20 S2EF-2M, where our method also achieves accuracy comparable to mainstream baselines.
>
> | Model | # Params (M) | Energy MAE (meV) | Force MAE (meV/Å) |
> | --- | --- | --- | --- |
> | GemNet-dT | 31 | 358 | 29.50 |
> | GemNet-OC | 38 | 286 | 25.70 |
> | SCN | 126 | 279 | 21.90 |
> | eSCN | 51 | 283 | 20.50 |
> | EquiformerV2 | 85 | 285 | 20.46 |
> | E2Former | 33 | 275 | 21.90 |
> | E2Former-V2 | 54 | 287 | 21.80 |

---

> > ### Author Rebuttal · Reviewer_bP3d · 2026-04-04
> >
> > > W2: The “FlashAttention” curve uses PyTorch scaled_dot_product_attention with a K-neighbor mask, so its theoretical complexity is O(kN). In practice, latency can grow faster due to masking overhead.
> >
> > In the abstract, you mentioned attention's complexity is O(N^2) while EGNN's is O(kN). Wouldn't that be the reason? Given these, I don't think putting them in the same figure is motivating.
> >
> > > W3 and Q2
> >
> > Equiformer paper compared dot product attention with their non-linear attention and showed the latter is better. The E2Former-V2 uses the former.
> >
> > Moreover, the question is about the value path — EquiformerV2 applies S² activation (or gate activation) to the value features, which is a non-linear equivariant operation that increases expressivity. E2Former-V2's value path is linear (Eq. 8-10 are all linear operations). Radial basis functions and attention weights are scalar nonlinearities, which both model have. The equivariant nonlinearity on higher-order edge features seems to be missing.
> >
> > > W8
> >
> > The results on SPICE and OMol25 are not comparable. E2Former-V2 has weaker performance. On SPICE, it is worse than eSEN. And it is unclear what is the sizes of models being compared. The comparison on training time with / without triton kernels should be included to show the model is both more accurate and faster. On OMol25, eSEN-sm is obviously better, not to mention the larger eSEN-md one, which is not included in the table. This raises the fundamental issue of whether using a weaker but faster model is actually helpful. The author should also address why accuracy is worse (e.g., no nonlinearity on edge features). It's ok to be less performant but faster, but it should be very clear where the trade-offs are made.
> >
> > > W12
> >
> > I respectfully disagree with the claimed contribution. Mostly it is the triton kernel that enable better efficiency. It is unclear whether the efficiency gained from the architecture would not affect its accuracy.
> >
> > > Q1
> >
> > Can you clarify that if triton kernel is removed, the compute / memory complexity is O(number of edges) or not?
> > Also please provide some detailed, self-contained explanation.
> >
> > > Q3
> >
> > The question is more like "to make this triton kernel, what changes you made to the architecture". The trade-offs in accuracy can be seen by comparing wiht eSEN on OMol25.
> >
> > > Q4
> >
> > I think the results on OC20 are not comparable in terms of energy and force MAE. First, E2Former-V2 has almost the same results to E2Former-V1. Thus, where are the differences? Second, number of parameters do not reflect training time. For example, eSCN is slower than EquiformerV2 at the same level of accuracy. Training time of E2Former-V2 should be reported with / without triton kernels to better reflect the gain. Third, the gap between E2Former-V2 and eSCN and EquiformerV2 is the same as the gap between SCN and eSCN and EquiformerV2. Forth, it is unclear if E2Former-V2 uses comparable hyper-parameters. For example, eSCN and EquiformerV2 are trained for 12 epochs. We can keep getting better results by training for more epochs.

---

> > > ### Author Response · Authors · 2026-04-07
> > >
> > > **Overall, we thank the reviewer for the detailed comments. We fully acknowledge the significant progress made by recent architectures—such as Equiformer, eSCN, EquiformerV2, eSEN, E2Former, UMA, and related works—which provide elegant mathematical formulations and achieve strong accuracy improvements.**
> > >
> > > **Our goal is complementary: rather than proposing a new expressive architecture, we focus on improving the practical efficiency and scalability of equivariant models. In particular, we aim to make equivcariant models feasible for large-scale molecular simulations by addressing the dominant runtime and memory bottlenecks. Therefore, some of the observed trade-offs (e.g., slightly lower accuracy in certain settings) reflect this design choice.**
> > >
> > > > W3 and Q2
> > > >
> > >
> > > Thank you for the insightful question. We apologize for the confusion. The non-linearity in the value path was not clearly described in the paper, which may have led to this misunderstanding. We will clarify this in the revised version.
> > >
> > > Regarding the value path, we would like to clarify that in our implementation (Supplementary Table 5, tp type = QKdotS_alpha + triton), the non-linearity is not removed but reorganized. Specifically, the feature is constructed as:
> > >
> > > Fea = nonlinear( (Q_i || V_j)^l ⊙ r_{ij}^l ),
> > >
> > > followed by attention weights:
> > > a_{ij} = softmax( vec2scalar(Q_i) * vec2scalar(K_j) * Fea ) * rbf .
> > >
> > > **In this sense, part of the non-linearity is shifted into the attention construction (including attention bias), rather than being applied explicitly on the value path**. We agree that this design choice was not clearly described in the paper, and we will add a more explicit explanation in the revision.
> > >
> > > Compared with EquiformerV2 and eSEN, their implementations apply SO(2) convolution → S² (or gate) activation → SO(2) convolution on edge features, which indeed introduces stronger equivariant non-linearity and increases expressivity. We agree with this observation.
> > >
> > > Our design instead trades explicit value-path equivariant non-linearity for a more efficient attention formulation, while still maintaining non-linearity through attention construction and radial features.
> > >
> > > > W8
> > > >
> > >
> > > Thank you for the detailed comments. On SPICE, eSEN achieves ~10% lower force MAE than E2Former-V2 in the conservative setting, while E2Former-V2 is ~10% better than eSEN-direct in the direct setting. On OMol25, the gap is within ~2% in the conservative setting, and E2Former-V2-direct is slightly better than eSEN-small-direct. We agree that eSEN-small (conservative) is the strongest reported accuracy.
> > >
> > > We followed results from the eSEN paper Table 1, where model sizes are not clearly specified; we will clarify this in the revision for a fairer comparison.
> > >
> > > Regarding “This raises the fundamental issue of whether using a weaker but faster model is actually helpful.”, our goal is scalability with competitive accuracy. In large-scale MD, E2Former-V2 reproduces experimental RDF in water (2 ns) and enables stable 2 ns simulation on a ~30k-atom protein, **demonstrating practical usefulness despite small accuracy gaps**.
> > >
> > > > Q1
> > > >
> > >
> > > Yes—when the Triton kernel is removed, **memory complexity become effectively O(E)**, where E ≈ kN is the number of edges, while the overall **computation remains O(Nk)**.
> > >
> > > | Aspect | Without Triton | With Triton |
> > > | --- | --- | --- |
> > > | Compute | O(Nk) | O(Nk) |
> > > | Memory | O(Nk) | O(N) |
> > > | Q/K/V | Materialized per edge | On-the-fly |
> > > | Bottleneck | Memory (HBM+allocation) | Compute |
> > >
> > > In a vanilla PyTorch implementation, attention is materialized at the edge level: for each edge (i, j), query, key, and value features need to be explicitly gathered and stored, resulting in intermediate tensors of size O(Nk).
> > >
> > > In contrast, with the Triton fused kernel, query/key/value features are computed on-the-fly for each edge and kept in GPU registers without materializing edge-level tensors. As a result, the memory cost for Q/K/V is reduced to O(N).
> > >
> > > Empirically, this difference is critical: without the Triton kernel, the vanilla implementation runs out of memory beyond ~5k atoms, whereas with the fused kernel we can scale to ~100k atoms.
> > >
> > > > Q3 and Q4
> > > >
> > >
> > > E2Former-V2 follows the same architectural design as E2Former, and no fundamental changes to the model architecture are introduced to enable the Triton kernel. Our goal is not to redesign the architecture, but to provide a scalable implementation that integrates algebraic sparsity with hardware-aware execution.
> > >
> > > In other words, the Triton kernel is designed to accelerate the existing computation pattern, rather than requiring architectural modifications. The architectural design itself is not the focus of this work; we refer readers to the E2Former paper for detailed discussion of the model design.

---

### Official Review · Reviewer_47sR · 2026-03-13

**Soundness:** 3
**Presentation:** 3
**Significance:** 3
**Originality:** 2
**Overall Recommendation:** 4
**Confidence:** 3

**Summary:**

This paper proposes E2Former-V2, an equivariant transformer architecture designed for efficient geometric deep learning. The work introduces Edge-Aggregated Attention Scheme and a fused on-the-fly attention kernel to reduce memory overhead and accelerate equivariant attention operations.

**Compliance With Llm Reviewing Policy:**

Affirmed.

**Key Questions For Authors:**

Do the authors plan to open-source the implementation? Releasing the code would significantly improve reproducibility and allow the community to better evaluate and adopt the proposed optimizations.

**Limitations:**

No. See weaknesses

**Strengths And Weaknesses:**

Strengths

1. Addresses efficiency challenges in equivariant transformers. The paper is well-motivated. It focuses on improving the efficiency of equivariant attention mechanisms, which is an important problem for scaling equivariant models.

2. Hardware-aware kernel optimization. The fused on-the-fly attention kernel can be helpful for other users in the community if it can be open-sourced.

3. Practical implementation insights. The implementation details are provided with well-designed kernel-level benchmarks.

Weaknesses

1. Absolute GPU utilization remains unclear. Although the paper reports substantial speedups over the PyTorch baseline and claims high hardware utilization, the absolute throughput values reported in Figure 4 appear to peak only at the order of tens of TFLOPS. In comparison, the NVIDIA H20 GPU provides roughly 148 TFLOPS FP16/BF16 Tensor Core throughput.

2. Lack of end-to-end component ablation. The paper provides microbenchmarks for the EAAS formulation and the fused attention kernel. However, it does not present a clear end-to-end ablation study isolating the contribution of each component to the final performance improvements. For example, it would be useful to compare: (1) the model without EAAS, (2) the model without the fused kernel implementation, and (3) the full proposed method.

3. Minor writing issues. The manuscript contains several small writing issues that should be corrected. For example, the phrase “tow datasets” appears instead of “two datasets”, and some sentences contain grammatical inconsistencies. There is also inconsistent notation between “E2FormerV2” and “E2Former-V2” in the text.

---

> ### Author Rebuttal · Authors · 2026-03-30
>
> Thank you for your detailed and constructive feedback. Below we provide detailed responses to your concerns. We will incorporate the corresponding experiments and clarifications in the revised paper.
>
> **W1:** Thank you for the comment. Our implementation uses FP32, whose theoretical peak on H20 is about 44 TFLOPS. In addition, our attention is sparse: each atom only attends to neighbors within the radial cutoff, which introduces irregular sparse indexing and memory access. Therefore, unlike dense matmul workloads, this kernel is not expected to achieve peak Tensor Core throughput, and the reported throughput is consistent with the nature of the computation.
>
> Our end-to-end ablation study also shows that the Triton fused kernel contributes more to the final speedup than EAAS. At the same time, there is still a clear gap between our current throughput and dense peak matmul performance as you mentioned. This suggests that further optimization may still be possible, and we plan to explore whether the sparse computation can be pushed closer to the hardware peak in future work.
>
> **W2:** Thank you for the suggestion. We agree that an end-to-end ablation isolating the contribution of each component is important, and we have now performed such a study on a conservative model setting on an H20 machine (average neighbor count ~ 50). We compare four configurations: **EAAS (with and without) and Triton fused kernel (with and without)**,
>
> | Atom Count | EAAS | Triton | Samples/s |
> | --- | --- | --- | --- |
> | 5000 | ✗ | ✗ | 1.40 |
> | 5000 | ✗ | ✓ | 3.31 |
> | 5000 | ✓ | ✗ | 1.55 |
> | 5000 | ✓ | ✓ | 4.24 |
> | 10000 | ✗ | ✗ | OOM |
> | 10000 | ✗ | ✓ | 1.65 |
> | 10000 | ✓ | ✗ | OOM |
> | 10000 | ✓ | ✓ | 2.12 |
>
> These results show that both components contribute, but in different ways. Since the edge-level computation is the primary bottleneck in both runtime and memory, the Triton fused kernel provides the dominant end-to-end gain. In contrast, enabling EAAS alone cannot remove the main edge-level memory bottleneck, which explains why configurations without the fused kernel still run into **OOM** at larger system sizes. The full system consistently achieves the best throughput, improving from **1.40 to 4.24 samples/s** at 5000 atoms. For throughput measurements with both EAAS and triton at larger atom counts, please also refer to **Table 3 (Inference Throughput Scaling)** in the paper.
>
> We will add this end-to-end ablation to the revised paper to make the contribution of each component clearer.
>
> **W3:** Thank you for pointing this out. We will correct these writing issues, including typos, grammar, and notation inconsistencies, in the revision.
>
> **Q: open-source the implementation?**
>
> Yes, we plan to open-source the implementation. The appendix already includes the tables and settings needed for reproduction. We also plan to extend the model to a distributed version for high-accuracy simulations at the scale of tens to hundreds of millions of atoms.

---

> > ### Author Rebuttal · Reviewer_47sR · 2026-04-03
> >
> > Thank the authors for the detailed rebuttal.
> >
> > Regarding the absolute GPU utilization, I would suggest including the corresponding absolute numbers along with additional analysis. It may also be helpful to discuss this aspect in the limitations section to further strengthen the work.

---

> > > ### Author Response · Authors · 2026-04-03
> > >
> > > Thank you for the suggestion. We have added absolute FLOPS measurements for dense FlashAttention on H20. Here, the feature dimension is head × D, and N denotes the number of atoms (or sequence length). Torch_SDPA refers to the PyTorch `scaled_dot_product_attention` function.
> > >
> > > We observe that when D = 8 (feature dim = 128), the two main attention matmuls (QKᵀ and AV) become highly skinny, leading to poor Tensor Core and SM utilization. In contrast, D = 64 better matches hardware-friendly tile sizes, enabling more efficient blocking, vectorization, and Tensor Core mapping, and thus significantly higher throughput. When the feature dimension reaches 1024, **dense FlashAttention approaches the FP32 peak of H20**.
> > >
> > > Overall, our sparse **Triton kernel is still about 10–20× slower than dense FlashAttention in compute-saturated regimes**. We agree this gap is important and will discuss it as a limitation and direction for future improvement in the revision.
> > >
> > > | N | head | D | Sparse_EGNN_TFLOPS | Torch_SDPA_Mask_TFLOPS | Torch_SDPA_Dense_TFLOPS |
> > > | --- | --- | --- | --- | --- | --- |
> > > | 128 | 16 | 8 | 0.015 | 0.051 | 0.191 |
> > > | 512 | 16 | 8 | 0.066 | 0.117 | 1.670 |
> > > | 2048 | 16 | 8 | 0.235 | 0.088 | 4.420 |
> > > | 8192 | 16 | 8 | 0.506 | 0.029 | 6.250 |
> > > | 32768 | 16 | 8 | 0.609 | 0.007 | 6.470 |
> > > | 128 | 16 | 64 | 0.110 | 0.345 | 1.330 |
> > > | 512 | 16 | 64 | 0.433 | 0.792 | 11.100 |
> > > | 2048 | 16 | 64 | 1.371 | 0.535 | 26.900 |
> > > | 8192 | 16 | 64 | 1.995 | 0.191 | 37.300 |
> > > | 32768 | 16 | 64 | 1.951 | 0.051 | 38.600 |

---

### Official Review · Reviewer_4D4q · 2026-03-16

**Soundness:** 3
**Presentation:** 3
**Significance:** 3
**Originality:** 3
**Overall Recommendation:** 5
**Confidence:** 4

**Summary:**

The authors combine E2Former’s binomial local expansion with eSCN’s axis-aligned technique to accelerate equivariant neural networks. And they implement the axis-alignment technique with Triton kernels after the local expansion to speed up the overall tensor product. Results on SPICE and OMol show their method significantly improves the throughput over prior equivariant networks.

**Compliance With Llm Reviewing Policy:**

Affirmed.

**Final Justification:**

The rebuttal has addressed my main concerns. The methodology is clean and useful, and the speed-up is remarkable, although the performance is not state-of-the-art.

**Key Questions For Authors:**

- In eSCN, an important bottleneck is not the SO(2)-convolution, but the Wigner D matrix multiplication of alignment rotations. What is the cost of that component in your network?

- This is more a question about E2Former itself. Converting edge complexity $\mathcal O(|\mathcal E|)$ to node complexity $\mathcal O(N)$ is a promising idea. However, if the target graph is very sparse, the number of edges is only a small constant factor of the number of nodes $\mathcal O(|\mathcal E|) = \mathcal O (kN)$ (as you write in Section 1), and $k$ is very small, say $k \le 5$. In this case, would E2Former's convolution still provide a meaningful benefit?

**Limitations:**

I do not find a section that explicitly states the limitations. I suggest that the authors include one, since a clear limitation of this work is its performance.

**Strengths And Weaknesses:**

## Strengths

- The paper, especially the reindexing part, is clearly written and demonstrated, with sufficiently large-scale experiments for throughput comparison.

- The speedup reported in Table 3 is phenomenal.

## Weaknesses

- The results in the OMol dataset are not SOTA. But I do not think the lack of SOTA performance diminishes the value of the paper since the speed improvements are pretty good. Still, the authors should explain why the model underperforms and discuss possible improvements.

- For Figure 3, e3nn tensor product is a slow baseline in itself. A better comparison would be against E2Former’s convolution or eSCN’s SO(2) convolution.

- Please cite relevant acceleration papers [1] [2] [3] [4], since these methods/theories also target (different aspects of) acceleration. In particular, it would be better to compare with the throughput of EscAIP [4].

- Please use the same font in Figure (b) as in Figure (a)


[1] The Price of Freedom: Exploring Expressivity and Runtime Tradeoffs in Equivariant Tensor Products. Yuqing Xie et al.

[2] Enabling Efficient Equivariant Operations in the Fourier Basis via Gaunt Tensor Products. Shengjie Luo et al.

[3] Tensor Decomposition Networks for Fast Machine Learning Interatomic Potential Computations. Yuchao Lin et al.

[4] The Importance of Being Scalable: Improving the Speed and Accuracy of Neural Network Interatomic Potentials Across Chemical Domains. Eric Qu et al.

---

> ### Author Rebuttal · Authors · 2026-03-30
>
> Thank you for your valuable feedback. The suggested experiments are very helpful, and we will include them in the revised paper.
>
>
> **W1:** Thank you for the question. We agree that E2Former-V2 is not the absolute SOTA on OMol, but it remains competitive with strong equivariant baselines while providing significant speedup. Possible improvements include:
>
> - using MoE to increase model capacity more efficiently,
> - designing stronger attention with more expressive queries and keys,
> - increasing the overall model size,
> - training on more data,
> - extending the number of training epochs,
> - and better combining different tensor-product types, such as (h_j  ⊗ r_{ij}^{0}), (h_j ⊗ r_{ij}^{1}), and (h_j ⊗ r_{ij}^{2}).
>
> **W2:** We agree that e3nn alone is not a sufficient baseline and will add comparisons with cuEquivariance and OpenEquivariance. For the regime feature = (h×0e + ... + h×l e) ⊗ SH = (1×l2e), SO2-EAAS is the fastest, followed by cuEquivariance, then e3nn, and then OpenEquivariance.
>
> (128x0e + 128x1e + 128x2e) ⊗ (1x2e), runtime (ms), (1e results omitted due to space)
>
> | method\number of⊗ | 1000 | 2000 | 4000 | 8000 | 16000 | 32000 |
> | --- | --- | --- | --- | --- | --- | --- |
> | so2 | 0.15 | 0.22 | 0.35 | 0.65 | 1.20 | 2.25 |
> | so3 e3nn | 0.88 | 0.90 | 1.06 | 1.51 | 2.39 | 4.25 |
> | so3 cuequivariance | 1.38 | 1.41 | 1.40 | 1.44 | 1.93 | 3.06 |
> | so3 openequivariance | 0.55 | 0.68 | 1.12 | 1.86 | 3.31 | 6.20 |
>
> **W3:** Thank you for these references. We will add [1]–[4] to related work and include EscAIP throughput in our comparison (direct-force setting):
>
> | Method | Inference Speed (samples/s) |
> | --- | --- |
> | GotenNet | 14.25 |
> | Allegro | 4.14 |
> | Equiformer-v2 | 6.69 |
> | EscAIP-S | 16.00 |
> | E2Former-V2 | 58.33 |
>
> **W4:** We will unify the font in Figure (b) with that in Figure (a).
>
> **Q1:** Thank you for this important question. At a high level, E2Former-V2 is an atom-level method with complexity O(N), while eSCN is an edge-level method with complexity O(E). We provide a more detailed analysis of neighbor dependence in the next answer.
>
> For E2Former-V2, the tensor product (h × 0e + h × 1e + ... + h × l e) ⊗ (1 × l2 e) leads to:
>
> - Wigner D transform: (l+1)^4 × h
> - EAAS reindex: (l+1)^2 × h
> - linear function: (l+1)^2 × h × h
>
> When h >> (l+1)^2, the linear term dominates; otherwise, the Wigner D transform is more significant.
>
> By contrast, eSCN uses (h × 0e + ... + h × l e) ⊗ (1 × 0e + ... + 1 × l e), with cost:
>
> - Wigner D transform: (l+1)^4 × h
> - SO(2) convolution (fully connected coupling): (l+1)^2 × l × h × h
>
> In practice, when h >> l, the SO(2) convolution dominates the edge-level cost; otherwise, the Wigner D term dominates.
>
> **Q2:** Thank you for the question. We agree that when the graph is very sparse (e.g., k = 2 or 5), the advantage of converting edge complexity to node complexity is limited.
>
> In E2Former, the node-level cost is O(N)(l+1)^2 h^2, while edge-level terms include attention O(kN)h, bias O(kN)h^2, and aggregation O(kN)(l+1)^2 h. For small k, the bottleneck is mainly node-level; as k increases, edge-level terms become dominant.
>
> E2Former inference speed for a 10,000-atom system
>
> | neighbor | runtime (ms) / step | sample per second |
> | --- | --- | --- |
> | 2 | 85.4 | 11.7 |
> | 5 | 89.4 | 11.2 |
> | 10 | 96.1 | 10.4 |
> | 20 | 118.3 | 8.4 |
> | 50 | 184.9 | 5.4 |
> | 100 | 282.0 | 3.5 |
>
> For edge-centric methods, the dominant cost is edge-level linear/nonlinear computation at least O(kN)(l+1)^2 h^2, so the runtime always depends much more strongly on k. This is also confirmed by our empirical results:
>
> Edge centric method
>
> | neighbor | runtime (ms)/ step | sample per second |
> | --- | --- | --- |
> | 2 | 70.4 | 14.2 |
> | 5 | 121.9 | 8.2 |
> | 10 | 212.2 | 4.7 |
> | 20 | 398.1 | 2.5 |
> | 50 | 962.9 | 1.0 |
> | 100 | 1872.7 | 0.5 |
>
> Overall, the speed gap is small when k is very low, but becomes significant once k reaches around 10 or above. In practical MLIP settings, k is typically about **50–100** for liquid water and **30–50** for OC20/OC22 under a 5–6Å cutoff.
>
> **Limitations.**
>
> A first limitation is representation flexibility. For tasks that do not require strict equivariance, non-equivariant models (e.g., EscAIP) may achieve stronger predictive performance. In contrast, for long-horizon physical simulations, equivariant models such as E2Former may still have advantages, since non-equivariant models can exhibit non-smooth and physically inconsistent behavior, as discussed in eSEN. We will further explore how to improve the predictive performance of E2Former.
>
> A second limitation is numerical precision. E2Former relies on relative coordinates (e.g., rij = ri - rj), which can introduce numerical errors under finite-precision computation. As a result, there is a potential numerical stability risk in large-scale or sensitive simulations.

---

> > ### Author Rebuttal · Reviewer_4D4q · 2026-03-31
> >
> > Thank you for your rebuttal. I believe this solves all my concerns, and I have raised my score.

---

### Decision · Program_Chairs · 2026-04-30

**Decision:**

Accept (regular)

**Comment:**

This paper improves the scalability of equivariant transformers for modeling 3D atomisitic systems, based on previous work which reduces from SO(3) to SO(2) convolution, and more importantly a hardware aware implementation of the attention mechanism which avoids explicitly computing a feature per edge. The proposed transformer is competitive on standard benchmarks, with a tradeoff between somewhat lower accuracy on the one hand, and $\times 3$ faster inference on the other hand. A more notable achievement is that this transformer can address simulations with 100K atoms, while competing methods run out of memory around 10K atoms. This last improvement seems subtantial and important to me, and in the opinion of most reviewers, and my opintion, merits acceptance.

 I'd like to stress to the authors the importance of making their code available and easy to use, especially due to the nature of the contribution of this paper. Also for the camera ready version please address issues of presentation and transparency raised by reviewer bP3d.